# Pivot Clustering to Minimize Error in Forecasting Aggregated Demand Streams Each Following an Autoregressive Moving Average Model

Vladimir Kovtun [1,*], Avi Giloni [2], Clifford Hurvich [3] and Sridhar Seshadri [4]

[1] Sy Syms School of Business, Yeshiva University, Suite 334, 215 Lexington Avenue, New York, NY 10012, USA
[2] Sy Syms School of Business, Yeshiva University, BH-428, 500 West 185th St., New York, NY 10033, USA; agiloni@yu.edu
[3] Technology, Operations and Statistics, Leonard N. Stern School of Business, New York University, 44 West 4th St., New York, NY 10012, USA; churvich@stern.nyu.edu
[4] Gies College of Business and Carle Illinois College of Medicine, University of Illinois at Urbana-Champaign, 601 E John Street, Champaign, IL 61820, USA; sridhar@illinois.edu
* Correspondence: vladimir.kovtun@yu.edu

**Abstract:** In this paper, we compare the effects of forecasting demand using individual (disaggregated) components versus first aggregating the components either fully or into several clusters. Demand streams are assumed to follow autoregressive moving average (ARMA) processes. Using individual demand streams will always lead to a superior forecast compared to any aggregates; however, we show that if several aggregated clusters are formed in a structured manner, then these subaggregated clusters will lead to a forecast with minimal increase in mean-squared forecast error. We show this result based on theoretical MSFE obtained directly from the models generating the clusters as well as estimated MSFE obtained directly from simulated demand observations. We suggest a pivot algorithm, which we call Pivot Clustering, to create these clusters. We also provide theoretical results to investigate sub-aggregation, including for special cases, such as aggregating demand generated by MA(1) models and aggregating demand generated by ARMA models with similar or the same parameters.

**Keywords:** forecasting aggregate demand; clustering time series; Pivot Clustering; ARMA model; order-up-to policy





## 1. Introduction

Modern-day technologies not only permit firms to accurately track their point of sales data and lost sales (purchases not made by customers due to a lack of inventory) data but also gather more granular data. These data streams deluge firms with information which can be either aggregated for planning purposes or considered in its entirety or follow an in-between approach. In this paper we analyze a model in which a retailer is faced with exactly the same choices and provide guidelines for combining the data for the purpose of forecasting demand.

Consider a retailer who has access to its individual customers' demand streams. Assume that each of these demand streams follow an ARMA model having possibly contemporaneously correlated shock sequences. The primary contribution of this research is to quantify to what extent such a retailer would benefit from forecasting each of the individual streams as opposed to the aggregate. In general, retailers forecast their aggregate demand stream since historically the retailer may only have accurate aggregate demand information, and the forecasting of the individual customer demand streams is often thought of as being cumbersome and time consuming. We demonstrate that a retailer observing multiple demand streams generated by ARMA models can drastically reduce its mean squared forecasting error (MSFE) by forecasting the individual demand streams as opposed to just the aggregate demand stream, as noted in [1].

Although the retailer's MSFE is never lower when forecasting using aggregated demand compared to the individual demand streams (this holds for the theoretical case when model coefficients are known and need not be estimated), there are cases when the individual streams do not reduce the MSFE. The primary situation where this occurs, as is discussed in the paper, is when various models generating the demand streams have identical ARMA parameter values. For the in-between case, our results demonstrate that the retailer's MSFE under aggregate forecasting can be greatly reduced if the retailer forecasts various clusters of aggregated demand streams. We also show by example that clustering continues to perform well in the event that ARMA models are estimated for non-ARMA data.

In other words, retailers can make use of data mining and other clustering approaches in order to generate clusters of similar customers and their demand streams. We show that such clustering methods significantly enhance forecast accuracy. This is related to the study of telephone data in [2], where the authors concluded that subaggregated data can be effective for improving forecast accuracy compared to aggregate data. They also note that data are often aggregated to the level that forecasts are required. We describe here ways in which subaggregated clusters can be selected to minimize forecast error. Several examples are mentioned in [3] in the context of assigning demand allocation to different facilities.

Many researchers and practitioners focus on the need to determine clusters of similarly situated customers in order to create and provide customized and/or personalized products. This type of clustering is generally performed on specific characteristics that customers possess (see, for example, [4] and the references within). On the other hand, our focus is on forecasting demand for a product by a firm's customers, recognizing that these customers may have different preferences and hence differing demand. As we describe below, from the demand forecasting perspective, the information contained within the individual demand streams provides the optimal forecast (in terms of minimizing the MSFE and hence inventory related costs) for product demand. Nonetheless, there has been research on the use of clustering methods within a forecasting environment when customer demand data are high dimensional (see, for example, [5]).

As opposed to generating clusters based upon customer preferences and customer demographics, we explain how clusters can be generated explicitly from the individual time series structure of the individual demand streams or customers. Even though it is always optimal from a forecasting perspective to use the individual streams, clusters of similar customer streams may be very helpful to the firm for other reasons as described above. Future empirical work would be necessary to investigate to what extent and in what contexts clusters generated based upon time series structure of the demand streams correlate to clusters based upon other customer preferences. In such a case where there exists such a relationship, firms could use clustering for simplifying their demand forecasting while identifying groups of customers to receive personalized products.

The purpose of this paper is to demonstrate that clustering based upon time series structure can be utilized within demand forecasting that is superior to forecasting aggregate demand and nearly as good as forecasting the individual demand streams. The structure of our paper is as follows. In the next section, we describe the demand framework and supply chain setting of our research problem as well as the way that theoretical MSFE computations are determined for the various forecasts (using aggregated demand processes) included herein. In Section 3, we illustrate (through example) that there exists a particular set of subaggregated clusters which results in an MSFE that is close to the MSFE obtained from using disaggregated streams and much lower than the MSFE obtained from the fully aggregated sequence. In Section 4, we describe how to cluster demand streams generated by ARMA models using Pivot Clustering and how this compares to other clustering methods. In Section 5, we describe an objective function that can be minimized to obtain an optimal assignment of streams to clusters in terms of MSFE reduction. Finally, we obtain theoretical results on how demand streams produced by MA(1) models can be clustered in the most efficient way possible to reduce the resulting subaggregated MSFE in Section 6.

## 2. Model Framework

We consider a retailer with possibly many large customers. In general, retailers forecast their aggregate demand stream since the forecasting of the individual customer demand streams is often thought of as being cumbersome and time consuming. We demonstrate that a retailer observing multiple streams of demand generated by ARMA models can drastically reduce its MSFE by forecasting the individual demand streams or aggregated clusters of similar demand streams. We limit the focus of this paper on ARMA models that describe stationary demand in order to keep the exposition as clear as possible. If we were to consider ARIMA (or seasonal ARIMA) models, then differencing (or seasonal differencing) would need to be carried out on the data to apply the methodology discussed here. We further note that even simple ARMA(1,1) models appearing in Section 4 can have coefficients that produce seasonal patterns in demand realizations.

Hence, consider a retailer that observes multiple demand streams for a single product $\{X_{1,t}\}, \{X_{2,t}\}, \ldots, \{X_{N,t}\}$. Each demand stream $\{X_k\}$ is assumed to be generated by an ARMA model with respect to a sequence of shocks $\{\epsilon_{k,t}\}$ given by

$$\Phi_k(B)X_{k,t} = \Theta_k(B)\epsilon_{k,t} \tag{1}$$

where $\Phi_k(z) = 1 + \Phi_{k,1}z + \ldots + \Phi_{k,p_k}z^{p_k}$ and $\Theta_k(z) = 1 + \Theta_{k,1}z + \ldots + \Theta_{k,q_k}z^{q_k}$, such that $\{X_{k,t}\}$ is invertible and causal with respect to $\{\epsilon_{k,t}\}$ (see Brockwell and Davis, page 77 for a definition and discussion about causality and invertibility). We denote the variance of each shock sequence $\sigma_k^2 = E[\epsilon_k^2]$. Furthermore, we note that the shock sequences are potentially contemporaneously correlated with $\sigma_{ij} = E[\epsilon_{i,t}\epsilon_{j,t}]$. In general, this set up guarantees that the shocks $\epsilon_{k,t}$ are the retailer's Wold shocks (see [6] pp. 187–188 for a description of a Wold decomposition of a time series) and that the MSFE of one-step-ahead leadtime demand (when using the disaggregated (individual) streams) is the sum of the elements in the covariance matrix $\Sigma_\epsilon$ where $\Sigma_{ij} = \sigma_{ij}$ such that $\sigma_k^2 = \sigma_{kk}$ (see Equation (7)).

The focus of this paper is evaluating the difference in one-step-ahead MSFEs at time $t$ when the forecast of leadtime demand, given by $\sum_{i=1}^{\ell+1}(X_{1,t+i} + X_{2,t+i} + \ldots + X_{N,t+i})$, is based on the different series described below where $C_{k,\tau} = X_{1,\tau}^{C_k} + \ldots + X_{n_{C_k},\tau}^{C_k}$. Studying one-step-ahead MSFEs is mathematically simpler than those for general leadtimes since the former does not depend on model parameters.

| | |
|---|---|
| Disaggregated (individual) sequences | $\{X_{1,\tau}\}_{\tau=-\infty}^{t}, \{X_{2,\tau}\}_{\tau=-\infty}^{t}, \ldots, \{X_{N,\tau}\}_{\tau=-\infty}^{t}$ |
| Subaggregated (clustered) sequences | $\{C_{1,\tau}\}_{\tau=-\infty}^{t}, \{C_{2,\tau}\}_{\tau=-\infty}^{t}, \ldots, \{C_{n,\tau}\}_{\tau=-\infty}^{t}$ |
| Aggregated (full) sequence | $\{D_\tau = X_{1,\tau} + X_{2,\tau} + \ldots + X_{N,\tau}\}_{\tau=-\infty}^{t}$ |

Our problem is related to the one posed by [7] where a two-stage supply chain was considered with the retailer observing two demand streams. The focus of that paper was in evaluating information sharing between the retailer and supplier in a situation where the retailer forecasts each demand stream separately. Here, we show the benefit to the retailer (the situation is really identical for any player in the supply chain that might be observing multiple demand streams, where information sharing does not take place) in determining the separate forecasts while considering the existence of (possibly) more than two demand streams. Kohn [1] was the first to identify conditions under which using the individual demand streams leads to a better forecast than using the aggregated sequence; however, he did not determine the MSFE in the two cases. The same conditions can be used to show that if streams are subaggregated into clusters where optimal clusters are always used, then the MSFE of the forecast decreases as the number of clusters increases. Our aim is to determine the optimal cluster assignment based on a predetermined choice of the number of clusters. The number of clusters can be based on the level of detailed data available to a firm or based on the tradeoff from lowering MSFE and increasing the complexity of the model when increasing the amount of clusters used.

In this paper, we extend the results of [7] by providing formulas for computing MSFEs under the possibility of more than two streams and a general leadtime in the situation where a player's (retailer's) forecast can only be based on their Wold shocks. We also describe how a retailer would forecast its demand by identifying clusters of similar demand streams and then forecasting each cluster after it is aggregated. We provide a pivot algorithm, which we call Pivot Clustering, for identifying a locally optimal assignment of streams into a fixed number of clusters. The algorithm will often find the best possible assignment. We also describe a fast clustering algorithm which results in a globally optimal assignment when demand streams are generated by independent MA(1) models. Our results show that after the algorithms are carried out, much of the benefit of forecasting individual streams can be obtained by forecasting the aggregated clusters.

*2.1. Forecasting Using the Disaggregated (Individual) Sequences $\{X_{1,\tau}\}_{\tau=-\infty}^{t}, \ldots, \{X_{n,\tau}\}_{\tau=-\infty}^{t}$*

In this section we are interested in forecasting $\sum_{i=1}^{\ell+1}(X_{1,t+i} + X_{2,t+i} + \ldots + X_{N,t+i})$ when the forecast is based on the diaggregated individual sequences  as discussed in the previous section. The forecasting contained here is a textbook multivariate time-series result along with the propagation described in [8] as seen in (5).   Thus, we consider  processes $\{X_{k,t}\}, \ldots, \{X_{N,t}\}$ such that $X_{k,t}$  generated by ARMA models given by

$$\Phi_k(B)X_{k,t} = \Theta_k(B)\epsilon_{k,t} \tag{2}$$

with $\sigma_{jk} = E[\epsilon_{j,t}\epsilon_{k,t}]$.

In this case, the disaggregated MSFE given by

$$MSFE_{ind} = E\left[\left(\sum_{k=1}^{N}\sum_{i=1}^{\ell+1} X_{k,t+i} - \sum_{k=1}^{N}\widehat{\sum_{i=1}^{\ell+1} X_{k,t+i}}\right)^2\right] \tag{3}$$

where $\sum_{i=1}^{\ell+1}\widehat{X_{k,t+i}}$ is the best linear forecast of leadtime demand at time $t$ for stream $\{X_{k,t}\}$ based on (2). We note that

$$E\left[\left(\sum_{i=1}^{\ell+1} X_{j,t+i} - \widehat{\sum_{i=1}^{\ell+1} X_{j,t+i}}\right)\left(\sum_{i=1}^{\ell+1} X_{k,t+i} - \widehat{\sum_{i=1}^{\ell+1} X_{k,t+i}}\right)\right] = \sigma_{jk}\sum_{i=0}^{\ell}\omega_{j,i}\omega_{k,i} \tag{4}$$

such that

$$\omega_{k,i} = \begin{cases} 0 & i < 0 \\ \psi_{k,i} & i = 0 \\ \omega_{k,i-1} + \psi_{k,i} & 0 < i < \ell+1 \\ \omega_{k,i-1} + \psi_{k,i} - \psi_{k,i-\ell-1} & i \geq \ell+1. \end{cases} \tag{5}$$

with $\psi_{k,i}$ is the $i$th coefficient appearing in the MA($\infty$) representation of $\{X_{k,t}\}$ with respect to $\{\epsilon_{k,t}\}$ (see [8] for details). That is, $1 + \psi_{k,1}z + \psi_{k,2}z^2 + \ldots = \Psi_k(z)$ and

$$X_{k,t} = \Psi_k(B)\epsilon_{k,t} \tag{6}$$

such that $\Psi_k(z) = \dfrac{\Theta_k(z)}{\Phi_k(z)}$.

Thus, the MSFE of the  best linear forecast (BLF) of leadtime demand when using the individual sequences

$$\{X_{1,\tau}\}_{\tau=-\infty}^{t}, \{X_{2,\tau}\}_{\tau=-\infty}^{t}, \ldots, \{X_{N,\tau}\}_{\tau=-\infty}^{t}$$

is given by

$$MSFEind = \sum_{k=1}^{N} \sum_{j=1}^{N} \sum_{i=0}^{\ell} \sigma_{kj} \omega_{j,i} \omega_{k,i}. \tag{7}$$

We note that the one-step-ahead ($\ell = 0$) disaggregated MSFE is given by $\sum_{j=1}^{N} \sum_{k=1}^{N} \sigma_{jk}$.

Thus, we can compare (7) with (26) below to determine the reduction in MSFE when using the individual sequences as opposed to the aggregated sequence.

### 2.2. ARMA Representation of a Summed Sequence $\{S_\tau = X_{1,\tau} + \ldots + X_{s,\tau}\}_{\tau=-\infty}^{t}$

In this subsection, we determine the ARMA representation of a series $\{S_t\}$ with respect to a series of Wold shocks, where $\{S_t\}$ is the sum of several ARMA-generated streams given by $\{S_t\} = \{X_{1,t} + X_{2,t} + \ldots + X_{s,t}\}$. Furthermore, we determine the variance of the Wold shocks appearing in this representation. This will allow us to determine the MSFE when forecasts are based on the fully aggregated series as well as when the forecasts are based on subaggregated clusters. In order to obtain the ARMA representation, we first need to obtain the spectral density $f_S(\lambda)$ and the covariance generating function $G_S(z)$ of $\{S_t\}$.

**Proposition 1.** *Let $\{S_t\} = \{X_{1,t} + X_{2,t} + \ldots + X_{s,t}\}$. The spectral density of $\{S_t\}$ is given by*

$$f_S(\lambda) = \sum_{i=1}^{s} f_{X_i}(\lambda) + \sum_{i=1}^{s-1} \sum_{j=i+1}^{s} \left( f_{X_i X_j}(\lambda) + \bar{f}_{X_i X_j}(\lambda) \right) \tag{8}$$

*where the cross-spectrum $f_{X_i X_j}(\lambda)$ is defined $f_{X_i, X_j}(\lambda) = \dfrac{1}{2\pi} \sum_{r=-\infty}^{\infty} e^{-i\lambda r} C_{X_i X_j}(r)$ with*

$C_{X_i X_j}(r) = E[X_{i,t+r} X_{j,t}]$ *and* $\bar{f}_{X_i X_j}(\lambda) = \dfrac{1}{2\pi} \sum_{r=-\infty}^{\infty} e^{i\lambda r} C_{X_i X_j}(r)$.

**Proof.** We will prove this by induction. Note that when $\{S_t\} = \{X_{1,t} + X_{2,t}\}$, $f_S(\lambda)$ is given by

$$
\begin{aligned}
f_S(\lambda) &= \frac{1}{2\pi} \sum_{r=-\infty}^{\infty} e^{-i\lambda r} E[(X_{1,t+r} + X_{2,t+r})(X_{1,t} + X_{2,t})] \tag{9} \\
&= \frac{1}{2\pi} \sum_{r=-\infty}^{\infty} e^{-i\lambda r} E[X_{1,t+r} X_{1,t}] + \frac{1}{2\pi} \sum_{r=-\infty}^{\infty} e^{-i\lambda r} E[X_{2,t+r} X_{2,t}] \\
&+ \frac{1}{2\pi} \sum_{r=-\infty}^{\infty} e^{-i\lambda r} E[X_{1,t+r} X_{2,t}] + \frac{1}{2\pi} \sum_{r=-\infty}^{\infty} e^{-i\lambda r} E[X_{2,t+r} X_{1,t}]. \tag{10}
\end{aligned}
$$

Noting that $\displaystyle\sum_{r=-\infty}^{\infty} e^{-i\lambda r} E[X_{2,t+r} X_{1,t}] = \sum_{r=-\infty}^{\infty} e^{i\lambda r} E[X_{2,t} X_{1,t+r}] = \bar{f}_{X_1 X_2}(\lambda)$, we see that

$$f_S(\lambda) = f_{X_1}(\lambda) + f_{X_2}(\lambda) + f_{X_1 X_2}(\lambda) + \bar{f}_{X_1 X_2}(\lambda)$$

which matches representation (8). $\quad\square$

Now suppose (8) holds for $\{S_{n,t}\} = \{X_{1,t} + \ldots + X_{n,t}\}$ processes. Thus,

$$f_{S_n}(\lambda) = \sum_{i=1}^{n} f_{X_i}(\lambda) + \sum_{i=1}^{n-1} \sum_{j=i+1}^{n} \left( f_{X_i X_j}(\lambda) + \bar{f}_{X_i X_j}(\lambda) \right) \tag{11}$$

Consider $\{S_{n+1}\} = \{S_{n,t} + X_{n+1,t}\}$. Since $\{S_{n,t}\}$ follows and ARMA model, we observe that

$$f_{S_{n+1}}(\lambda) = f_{S_n}(\lambda) + f_{X_{n+1}}(\lambda) + f_{S_n,X_{n+1}}(\lambda) + \bar{f}_{S_n,X_{n+1}}(\lambda) \tag{12}$$

Note that starting with the definition of $f_{S_n,X_{n+1}}(\lambda)$,

$$f_{S_n,X_{n+1}}(\lambda) = \frac{1}{2\pi} \sum_{r=-\infty}^{\infty} e^{-i\lambda r} E[S_{n,t}X_{n+1,t+r}] \tag{13}$$

$$= \frac{1}{2\pi} \sum_{r=-\infty}^{\infty} e^{-i\lambda r} E[X_{1,t}X_{n+1,t+r} + \ldots + X_{n,t}X_{n+1,t+r}] \tag{14}$$

$$= \frac{1}{2\pi} \sum_{r=-\infty}^{\infty} e^{-i\lambda r} E[X_{1,t}X_{n+1,t+r}] + \ldots + \frac{1}{2\pi} \sum_{r=-\infty}^{\infty} e^{-i\lambda r} E[X_{n,t}X_{n+1,t+r}] \tag{15}$$

$$= f_{X_1 X_{n+1}}(\lambda) + \ldots + f_{X_n X_{n+1}}(\lambda) \tag{16}$$

Similarly,

$$\bar{f}_{S_n,X_{n+1}}(\lambda) = \bar{f}_{X_1 X_{n+1}}(\lambda) + \ldots + \bar{f}_{X_n X_{n+1}}(\lambda)$$

Thus, from (12),

$$\begin{aligned} f_{S_{n+1}}(\lambda) &= f_{S_n}(\lambda) + f_{X_{n+1}}(\lambda) + f_{X_1 X_{n+1}}(\lambda) + \ldots \\ &+ f_{X_n X_{n+1}}(\lambda) + \bar{f}_{X_1 X_{n+1}}(\lambda) + \ldots + \bar{f}_{X_n X_{n+1}}(\lambda) \end{aligned} \tag{17}$$

or equivalently

$$\begin{aligned} f_{S_{n+1}}(\lambda) = \sum_{i=1}^{n} f_{X_i}(\lambda) &+ \sum_{i=1}^{n-1} \sum_{j=i+1}^{n} \left( f_{X_i X_j}(\lambda) + \bar{f}_{X_i X_j}(\lambda) \right) \\ &+ f_{X_{n+1}}(\lambda) + f_{X_1 X_{n+1}}(\lambda) + \ldots \\ &+ f_{X_n X_{n+1}}(\lambda) + \bar{f}_{X_1 X_{n+1}}(\lambda) + \ldots + \bar{f}_{X_n X_{n+1}}(\lambda) \end{aligned}$$

which can be simply written as

$$f_{S_{n+1}}(\lambda) = \sum_{i=1}^{n+1} f_{X_i} + \sum_{i=1}^{n} \sum_{j=i+1}^{n+1} \left( f_{X_i X_j}(\lambda) + \bar{f}_{X_i X_j}(\lambda) \right) \tag{18}$$

and the result is proved. □

Now, consider the covariance generating function $G_S(z) = \sum_{j=-\infty}^{\infty} E[S_t S_{t-j}] z^j$ of $\{S_t\}$.

Here, we use the equivalence $G_S(e^{-i\lambda}) = 2\pi f_S(\lambda)$ and note the following:

$$f_{X_i}(\lambda) = \frac{\sigma_i^2}{2\pi} \frac{|\Theta_i(e^{-i\lambda})|^2}{|\Phi_i(e^{-i\lambda})|^2} \tag{19}$$

$$f_{X_i X_j}(\lambda) = \frac{\sigma_{ij}}{2\pi} \frac{\Theta_i(e^{-i\lambda})}{\Phi_i(e^{-i\lambda})} \frac{\Theta_j(e^{i\lambda})}{\Phi_j(e^{i\lambda})} \tag{20}$$

$$\bar{f}_{X_i X_j}(\lambda) = \frac{\sigma_{ij}}{2\pi} \frac{\Theta_i(e^{i\lambda})}{\Phi_i(e^{i\lambda})} \frac{\Theta_j(e^{-i\lambda})}{\Phi_j(e^{-i\lambda})} \tag{21}$$

Thus, from Proposition 1 we observe that

$$G_S(z) = \sum_{i=1}^{s} \sigma_i^2 \frac{\Theta_i(z)\Theta_i(z^{-1})}{\Phi_i(z)\Phi_i(z^{-1})} + \sum_{i=1}^{s-1} \sum_{j=i+1}^{s} \left( \sigma_{ij} \frac{\Theta_i(z)}{\Phi_i(z)} \frac{\Theta_j(z^{-1})}{\Phi_j(z^{-1})} + \sigma_{ij} \frac{\Theta_i(z^{-1})}{\Phi_i(z^{-1})} \frac{\Theta_j(z)}{\Phi_j(z)} \right) \tag{22}$$

As described in Theorem 5 of [7], the covariance generating function $G_S(z)$ can be factorized as the ratio $\dfrac{O(z)P(z)}{Q(z)}$ where $O(z)$, $P(z)$ and $Q(z)$ are Laurent polynomials, with $O(z)$ having all its roots on the unit circle and $P(z)$ and $Q(z)$ having no roots on the unit circle. This result follows from the fact that each additive term in (76) is a ratio of Laurent polynomials and the fact that for any Laurent polynomial $P(z)$, both $P(z)P(z^{-1})$ and $P(z) + P(z^{-1})$ will be Laurent polynomials. Furthermore, if $P_1(z)$ and $P_2(z)$ are Laurent polynomials then $P_1(z)P_2(z)$ and $P_1(z) + P_2(z)$ will be Laurent polynomials as well.

We can now use the factorization provided in [9] and described in [7] to obtain the ARMA representation of $\{S_t\}$ with respect to the Wold shocks $\{\epsilon_t\}$ (appearing in its Wold representation). It should be noted that Remark 1 is simply a restatement of Theorem 5 of [7] with the slight addition of determining the polynomials appearing in the ARMA representation of the aggregate sequence $\{S_t\}$.

**Remark 1.** *$\{S_t\}$ can be represented with respect to shocks $\{\epsilon_t\}$ using the ARMA model*

$$\Phi(B)S_t = \Theta(B)\epsilon_t \tag{23}$$

*where $\Theta(z) = \prod_{i=1}^{m}(1 - a_i z)$ where $\{a_i\}$ are the roots of $O(z)P(z)$ on or inside the unit circle and*

$$\Phi(z) = \prod_{i=1}^{n}(1 - b_i z) \text{ where } \{b_i\} \text{ are the roots of } Q(z) \text{ inside the unit circle. Furthermore,}$$

$$\sigma_\epsilon^2 = E[\epsilon_t^2] = \frac{p_m \prod_{j=1}^{m}(-1/a_j)}{q_n \prod_{j=1}^{n}(-1/b_j)} \tag{24}$$

*where $p_m$ is the coefficient of $z^m$ in $P(z)$ and $q_n$ is the coefficient of $z^n$ in $Q(z)$.*

### 2.3. Forecasting Using the Fully Aggregated Sequence $\{D_\tau\}_{\tau=-\infty}^{t}$ for a General Leadtime

We note that Remark 1 can be used to obtain the ARMA representation of $\{D_t\}$ with respect to its Wold shocks $\{\epsilon_t\}$ as well as $\sigma_\epsilon^2 = E[\epsilon_t^2]$. We can therefore use Lemma 1 of [8] and its proof to determine the BLF of $\left(\sum_{k=1}^{\ell+1} D_{t+k}\right)$ and its MSFE when forecasting using the infinite past of $\{D_t\}$ up to time $t$, namely $\{D_\tau\}_{\tau=-\infty}^{t}$. That is, if we consider the MA($\infty$) representation of $\{D_t\}$ with respect to $\{\epsilon_t\}$ given by

$$D_t = \Psi(B)\epsilon_t \tag{25}$$

where $\Psi(z) = 1 + \Psi_1 z + \Psi_2 z^2 + \ldots$, then the MSFE of the BLF when using $\{D_\tau\}_{\tau=-\infty}^{t}$ is

$$MSFEagg = \sigma_\epsilon^2 \sum_{i=0}^{\ell} \omega_i^2 \tag{26}$$

where

$$\omega_i = \begin{cases} 0 & i < 0 \\ \psi_i & i = 0 \\ \omega_{i-1} + \psi_i & 0 < i < \ell + 1 \\ \omega_{i-1} + \psi_i - \psi_{i-\ell-1} & i \geq \ell + 1. \end{cases} \tag{27}$$

We note that $\Psi(z) = \dfrac{\Theta(z)}{\Phi(z)}$.

### 2.4. Forecasting Using Subaggregated Sequences

In this section we use the results in the previous section to create a forecast and compute its MSFE when some of the individual streams are subaggregated. That is, for $k \in 1 \ldots n$, let cluster $C_k$ consist of $n_{C_k}$ streams such that $\{C_{k,t} = X_{1,t}^{C_k} + \ldots + X_{n_{C_k},t}^{C_k}\}$. We are interested in the forecast and MSFE based on $\{C_{1,\tau}\}_{\tau=-\infty}^{t}, \ldots, \{C_{n,\tau}\}_{\tau=-\infty}^{t}$.

Section 2.2 describes how we can obtain the ARMA representation and variance of the Wold shocks appearing in the Wold representation for each sequence $\{C_{k,t}\}$. This can then be used to create a forecast from each subaggregated sequence, the sum of which can be taken as the forecast for $D_{t+\ell+1}$. The one-step ahead MSFE of this forecast is the sum of the entries of the covariance matrix of the Wold shocks appearing in the Wold representation of $\{C_{1,t}, \ldots, C_{n,t}\}$. Equation (7) describes how we can also use the covariance matrix of the shocks of $\{C_{1,t}, \ldots, C_{n,t}\}$ to obtain the MSFE for multi-step ahead forecasts. The remainder of this section will focus on obtaining this covariance matrix.

Without loss of generality, consider two subaggregated series $\{C_{1,t} = X_{1,t} + \ldots + X_{a,t}\}$ and $\{C_{2,t} = X_{a+1,t} + \ldots + X_{b,t}\}$ with ARMA representations

$$\phi_1^\star(B)C_{1,t} = \theta_1^\star(B)\epsilon_{1,t}^\star \tag{28}$$
$$\phi_2^\star(B)C_{2,t} = \theta_2^\star(B)\epsilon_{2,t}^\star \tag{29}$$

where $\{\epsilon_{1,t}^\star\}$ and $\{\epsilon_{2,t}^\star\}$ are the shocks appearing in the Wold representation of $\{C_{1,t}\}$ and $\{C_{2,t}\}$, respectively. We note that the variances of $\{\epsilon_{1,t}^\star\}$ and $\{\epsilon_{2,t}^\star\}$ can be obtained using Remark 1. To obtain the covariance $\sigma_{12}^\star = E[\epsilon_{1,t}^\star \epsilon_{2,t}^\star]$ consider the following.

We can rewrite the ARMA representations above as

$$\epsilon_{1,t}^\star = \frac{\phi_1^\star(B)}{\theta_1^\star(B)}C_{1,t} \tag{30}$$

$$\epsilon_{2,t}^\star = \frac{\phi_2^\star(B)}{\theta_2^\star(B)}C_{2,t} \tag{31}$$

The ARMA representations of $\{X_{1,t}\}, \ldots, \{X_{b,t}\}$ can also be rewritten as

$$X_{1,t} = \frac{\Theta_1(B)}{\Phi_1(B)}\epsilon_{1,t} \quad \ldots \quad X_{b,t} = \frac{\Theta_b(B)}{\Phi_b(B)}\epsilon_{b,t} \tag{32}$$

Based on the definition of $C_{1,t} = X_{1,t} + \ldots + X_{a,t}$ and $C_{2,t} = X_{a+1,t} + \ldots + X_{b,t}$ we observe that

$$\epsilon_{1,t}^\star = \frac{\phi_1^\star(B)}{\theta_1^\star(B)}\left[\frac{\Theta_1(B)}{\Phi_1(B)}\epsilon_{1,t} + \ldots + \frac{\Theta_a(B)}{\Phi_a(B)}\epsilon_{a,t}\right] \tag{33}$$

$$\epsilon_{2,t}^\star = \frac{\phi_2^\star(B)}{\theta_2^\star(B)}\left[\frac{\Theta_{a+1}(B)}{\Phi_{a+1}(B)}\epsilon_{a+1,t} + \ldots + \frac{\Theta_b(B)}{\Phi_b(B)}\epsilon_{b,t}\right] \tag{34}$$

To obtain $E[\epsilon_{1,t}^\star \epsilon_{2,t}^\star]$ we need to compute the expectation of the the product of the right-hand-sides of these two equations. Thus, we need to consider the sum of terms such as

$$E\left[\frac{\phi_1^\star(B)}{\theta_1^\star(B)}\frac{\Theta_i(B)}{\Phi_i(B)}\epsilon_{i,t}\frac{\phi_2^\star(B)}{\theta_2^\star(B)}\frac{\Theta_j(B)}{\Phi_j(B)}\epsilon_{j,t}\right]. \tag{35}$$

Note that we can write

$$\frac{\phi_1^\star(B)}{\theta_1^\star(B)}\frac{\Theta_i(B)}{\Phi_i(B)}\epsilon_{i,t} = \sum_{k=0}^{\infty}\tilde{\psi}_{i,k}\epsilon_{i,t-k} \tag{36}$$

and

$$\frac{\phi_2^\star(B)}{\theta_2^\star(B)}\frac{\Theta_j(B)}{\Phi_j(B)}\epsilon_{j,t} = \sum_{k=0}^{\infty}\tilde{\psi}_{j,k}\epsilon_{j,t-k} \tag{37}$$

where $\tilde{\psi}_{i,k}$ and $\tilde{\psi}_{j,k}$ can be obtained in the same way as the MA($\infty$) coefficients in (27). Hence, the term in Equation (35) can be rewritten as

$$E\left[\sum_{k=0}^{\infty}\tilde{\psi}_{i,k}\epsilon_{i,t-k}\sum_{k=0}^{\infty}\tilde{\psi}_{j,k}\epsilon_{j,t-k}\right] \tag{38}$$

Since the shock sequences are not correlated across time by assumption, this is equivalent to

$$E\left[\sum_{k=0}^{\infty}\tilde{\psi}_{i,k}\tilde{\psi}_{j,k}\epsilon_{i,t-k}\epsilon_{j,t-k}\right] \tag{39}$$

or equivalently

$$\sum_{k=0}^{\infty}\tilde{\psi}_{i,k}\tilde{\psi}_{j,k}\sigma_{ij} \tag{40}$$

Adding up these terms as required would yield the covariance. Hence

$$\sigma_{12}^{\star} = E[\epsilon_{1,t}^{\star}\epsilon_{2,t}^{\star}] = \sum_{i=1}^{a}\sum_{j=a+1}^{b}\sum_{k=0}^{\infty}\tilde{\psi}_{i,k}\tilde{\psi}_{j,k}\sigma_{ij} \tag{41}$$

We note that this methodology can easily be extended to obtain any of the covariances in the covariance matrix

$$\Sigma_{\epsilon}^{\star} = \begin{pmatrix} \sigma_{11}^{\star} & \cdots & \sigma_{1n}^{\star} \\ \sigma_{21}^{\star} & \ddots & \sigma_{2n}^{\star} \\ \vdots & & \vdots \\ \sigma_{n1}^{\star} & \cdots & \sigma_{nn}^{\star} \end{pmatrix} \tag{42}$$

We will use the previous methodology for all theoretical MSFE computations found in this paper. In the next subsection, we provide an example describing the importance of forecasting leadtime demand based upon the individual sequences.

*2.5. Example*

Consider a retailer that observes aggregate demand $\{D_t = X_{1,t} + X_{2,t} + X_{3,t}\}$ where each individual demand stream is generated by one of the following ARMA models:

$$X_{1,t} = (1 - 0.9B)\epsilon_{1,t} \tag{43}$$
$$X_{2,t} = (1 + 0.9B)\epsilon_{2,t} \tag{44}$$
$$X_{3,t} = (1 + 0.9B)\epsilon_{3,t} \tag{45}$$

where the shock covariance matrix is given by $\Sigma = \begin{pmatrix} 1.6 & -1.4 & 0.5 \\ -1.4 & 1.3 & -0.8 \\ 0.5 & -0.8 & 2.0 \end{pmatrix}$

We use the results described in Sections 2.1 and 2.2 to compare the MSFE of the forecasts of leadtime demand $\sum_{j=1}^{\ell+1} D_{t+j}$ when using the individual sequences $\{X_{1,t}\}_{\tau=-\infty}^{t}$, $\{X_{2,t}\}_{\tau=-\infty}^{t}$, $\{X_{3,t}\}_{\tau=-\infty}^{t}$ versus when using the aggregate sequence $\{D_{\tau}\}_{\tau=-\infty}^{t}$.

The covariance generating function of $\{D_t\}$ is given by

$$G_S(z) = \frac{0.090z^{-1} + 5.631 + 0.090z}{1} \tag{46}$$

and the ARMA representation of $\{D_t\}$ is given by

$$D_t = (1 + 0.01598704B)\epsilon_t \tag{47}$$

where $\sigma_\epsilon^2 = 5.629561$. We can check that the ARMA representation of $\{D_t\}$ and its covariance generating function match up by noting that

$$
\begin{aligned}
G_S(z) &= \sigma_\epsilon^2 \frac{\Theta(z)\Theta(z^{-1})}{\Phi(z)\Phi(z^{-1})} \\
&= 5.629561(1 + 0.01598704z)(1 + 0.01598704z^{-1}) \\
&= 0.090z^{-1} + 5.631 + 0.090z \tag{48}
\end{aligned}
$$

with $\Theta(z) = (1 + 0.01598704z)$ and $\Phi(z) = 1$ being the MA and AR polynomials appearing in the ARMA representation of $\{D_t\}$ as defined in Remark 1.

We note that the one-step ahead forecast then has a MSFE = 5.629561 and a two-step-ahead forecast (of $D_{t+1} + D_{t+2}$) has an MSFE = 11.44056.

Using Equation (7) and noting that $\omega_{k,0} = \psi_{k,0} = 1$ in Equation (5), we can obtain the MSFE of the forecast based on the individual demand streams. In this case, the MSFE is 1.5, which is 4.129561 (73.3%) lower that the forecast error when using the aggregated series. We can likewise compute the elements $\omega_{k,i}$ when forecasting $D_{t+1} + D_{t+2}$. In this case, $\omega_{k,0} = \psi_{k,0} = 1$ and $\omega_{1,1} = 0.1$, $\omega_{2,1} = 1.9$, and $\omega_{3,1} = 1.9$. From (7) this implies that the MSFE = 7.311, which is also 4.129561 (36.1%) lower (In this case, the reduction in MSFE appears the same, regardless of leadtime; however, this is due to the series being generated by MA(1) models. For higher order ARMA models, the reduction in MSFE may be dependent on leadtime.) than the forecast error when using the aggregated series. In the next section we demonstrate, with example, the benefit of forecasting using subaggregated clusters (which would have been identified using a suitable technique).

## 3. The Benefit of Forecasting Using Subaggregated Clusters

In this section, we consider a retailer that has ten demand streams that aggregate into three clusters consisting of similar streams. The models generating these streams are specifically chosen to provide a clear separation between "good" clusters leading to low MSFE and "bad" clusters leading to high MSFE. Intuition gleaned from Sections 6 and 7 hint that streams generated from ARMA models with similar coefficients should be clustered together. Hence, for our example we consider three groups of models having similar sets of coefficients within each group. Later, in Section 4, we will randomly assign coefficients to streams and still observe a sharp drop in MSFE when streams are clustered to minimize MSFE.

Suppose the retailer observes aggregate demand $\{D_t = X_{1,t} + X_{2,t} + \ldots + X_{10,t}\}$ where each individual demand stream is generated by one of the following ARMA processes:

$$
\begin{aligned}
(1 - 0.3B - 0.6B^2)X_{1,t} &= (1 - 0.6B - 0.2B^2)\epsilon_{1,t} \tag{49} \\
(1 - 0.35B - 0.5B^2)X_{2,t} &= (1 - 0.65B - 0.15B^2)\epsilon_{2,t} \tag{50} \\
(1 - 0.27B - 0.55B^2)X_{3,t} &= (1 - 0.63B - 0.17B^2)\epsilon_{3,t} \tag{51} \\
(1 - 0.8B)X_{4,t} &= \epsilon_{4,t} \tag{52} \\
(1 - 0.9B)X_{5,t} &= \epsilon_{5,t} \tag{53} \\
(1 - 0.75B)X_{6,t} &= \epsilon_{6,t} \tag{54} \\
(1 + 0.77B)X_{7,t} &= (1 + 0.6B)\epsilon_{7,t} \tag{55} \\
(1 + 0.68B)X_{8,t} &= (1 + 0.55B)\epsilon_{8,t} \tag{56} \\
(1 + 0.73B)X_{9,t} &= (1 + 0.52B)\epsilon_{9,t} \tag{57} \\
(1 + 0.7B)X_{10,t} &= (1 + 0.5B)\epsilon_{10,t} \tag{58}
\end{aligned}
$$

where the shock covariance matrix is given by

$$
\Sigma = \begin{pmatrix}
2 & 1 & 0.8 & -0.9 & -1.2 & -1.5 & 0.8 & 0.9 & 0.95 & 1 \\
1 & 2.1 & 0.7 & -0.6 & -0.5 & -0.4 & 0.21 & 0.31 & 0.36 & 0.39 \\
0.8 & 0.7 & 2.2 & -0.5 & -1.3 & -1 & 0.4 & 0.8 & 1 & 1.1 \\
-0.9 & -0.6 & -0.5 & 3 & 1.8 & 1.9 & -2 & -2.1 & -2.2 & -2.3 \\
-1.2 & -0.5 & -1.3 & 1.8 & 3.2 & 2 & -1.9 & -1.8 & -1.7 & -1.5 \\
-1.5 & -0.4 & -1 & 1.9 & 2 & 3.3 & -2.2 & -2.3 & -2.4 & -2.5 \\
0.8 & 0.21 & 0.4 & -2 & -1.9 & -2.2 & 5 & 1 & 1.25 & 1.5 \\
0.9 & 0.31 & 0.8 & -2.1 & -1.8 & -2.3 & 1 & 5.1 & 1.3 & 1.6 \\
0.95 & 0.36 & 1 & -2.2 & -1.7 & -2.4 & 1.25 & 1.3 & 5.7 & 1.8 \\
1 & 0.39 & 1.1 & -2.3 & -1.5 & -2.5 & 1.5 & 1.6 & 1.8 & 5.9
\end{pmatrix}
\tag{59}
$$

Consider the three natural clusters in the above ten demand streams, namely, streams 1–3, 4–6, and 7–10. It can be shown that, indeed, this grouping is optimal out of any other possible choice of three clusters simply by looking at all possible combinations and computing their MSFEs. Our analysis of the MA(1) case in Section 6 also points to these being the correct clusters based on the proximity of the ARMA coefficients of the models generating the demand streams. We demonstrate that, even though the best the retailer can achieve in such a situation is forecast all ten streams individually, if the retailer correctly clusters the demand streams as mentioned, the results will be similar. All MSFEs stated in this section are obtained using the methods described in Section 2.

Specifically in this case, if the retailer were to forecast the individual streams, its one-step ahead theoretical MSFE would be 21.64. If the retailer were to consider the subaggregated processes consisting of the correct clusters and forecast these separately, then the MSFE would be 21.74. This is in stark contrast to an MSFE of 61.39 when forecasting using the aggregate process of all ten streams. In other words, it is sufficient to determine clusters of similar customers as opposed to forecasting each individual stream in order to keep inventory-related costs down.

To see the impact of choosing the correct clusters, we consider the case that the retailer incorrectly clusters the streams as 1–2, 3–5, and 6–10. In this case, the MSFE rises to 33.4. Similarly, if the clusters chosen are 1&4, 2&3&5, and 6–10, then the MSFE is 45.04. Assigning streams randomly to three clusters consisting of three, two, and five streams yields Table 1.

**Table 1.** The MSFEs for various clusters of three, two, and five streams.

| MSFE | Clusters |
| --- | --- |
| 52.34495576 | 6,10,9 and 1,2 and 8,3,4,5,7 |
| 51.90912188 | 2,10,5 and 3,8 and 4,1,7,6,9 |
| 31.40789218 | 7,2,10 and 8,9 and 5,4,6,3,1 |
| 44.15962369 | 10,1,7 and 3,4 and 8,2,9,6,5 |
| 50.32525078 | 5,8,3 and 1,10 and 4,2,7,6,9 |
| 39.31100769 | 6,9,5 and 10,7 and 1,3,8,4,2 |
| 45.09358141 | 3,1,10 and 5,8 and 2,4,9,7,6 |
| 51.54828609 | 9,5,10 and 4,2 and 7,6,8,1,3 |
| 34.21154829 | 8,7,2 and 1,9 and 6,4,5,10,3 |
| 55.21445794 | 6,1,10 and 2,4 and 7,3,8,5,9 |

We note that there could be substantial reduction in MSFE even when multiple streams are clustered incorrectly. In the next section, we demonstrate how a retailer would be able to generate clusters of its individual demand streams using Pivot Clustering.

## 4. TS Clustering Algorithms and Pivot Clustering; Empirical Evaluation

We have shown that the when the retailer forecasts using clusters of its demand streams as opposed to the individual streams, its MSFE can vary greatly from close to the optimal value (when forecasting using the individual streams) to close to the MSFE of a

forecast based on the aggregate of all the streams. Hence, if the retailer wishes to forecast using clusters of its demand streams, the selection of the clusters is important. In general, a retailer might have customer-related information that could be used to generate the clusters. The benefit of generating the clusters and forecasting them is that there could be situations where it would be cumbersome for the retailer to collect the individual demand streams and service them individually. Forecasting clusters would therefore be a second-best option.

We propose Pivot Clustering for determining clusters which usually result in a relatively low MSFE among all choices of streams to clusters. We consider two ways to obtain the subaggregated MSFE based on some clustering assignment. The first is to use the individual ARMA demand models appearing in (2) to compute the subaggregated theoretical MSFE as per Sections 2.2–2.4. We note that if there is only one cluster, then the MSFE is the one computed for a forecast based on the aggregate of the all the streams, while if the number of clusters is equal to the number of streams, the MSFE is for a forecast based on the disaggregated (individual) sequences. We also estimate the MSFE by generating demand realizations for each stream based on (2). Once demand realizations are simulated for each stream, we subaggregate the realizations based on our choice of clusters. So, if some cluster is made up of streams $\{X_{i,t}\}$ and $\{X_{j,t}\}$, we say that the cluster has realization $\{X_{i,t} + X_{j,t}\}$. We then estimate an ARMA(5,5) (the AR and MA degrees were chosen with the understanding that these degrees increase with the number of streams subaggregated into a particular cluster while trying to limit the complexity of the models being estimated) model using each cluster's realization. Finally, we use the estimated models to obtain in-sample forecast errors and compute the covariance matrix of these forecast errors to estimate the MSFE for a particular assignment of streams to clusters. In the analysis below we see that the estimated MSFEs are close to theoretical ones and often lead to similar choices of clusters. Based on a predetermined number of clusters $n$, Pivot Clustering works as follows.

For each stream, randomly assign it to a cluster.

For each cluster:

For each stream in the cluster, compute or estimate the MSFE for the current assignment along with the resulting MSFEs if the stream was moved to each of the other clusters.

The MSFE can be either estimated based on realizations of the demand streams or computed using Equations (28), (29), (41) and (42).

Move each stream in the cluster to a cluster which leads to the largest overall MSFE reduction among all choices of clusters.

In the remainder of this section, we perform various simulations to assess the efficacy of Pivot Clustering. We focus on ARMA(1,1) models as these do not require too much runtime for Pivot Clustering based on theoretical MSFE and are complex enough to describe demand data such as in [10]. Additionally, forecasting an aggregate of ARMA(1,1) demand sequence has been studied by [11], where forecasts were based on exponential smoothing. The methods herein are generally applicable, however, to higher-order ARMA models. From a computational standpoint, it is possible to determine theoretical MSFE based on the aggregate of up to twenty demand streams generated by ARMA(1,1) models. The burden lies in having to find roots of large degree polynomials in order to determine the ARMA model and shock variance that describes the aggregated sequence. To understand the computational requirements, we check the runtimes of Pivot Clustering when determining theoretical and estimated MSFEs. Based on a given number of streams (between 10 and 20), we carry out Pivot Clustering for twenty different combinations of ARMA(1,1) models and plot the average of the runtimes in Figure 1 in assigning the streams to four clusters. If using estimated MSFE, then many more streams can be clustered, and Pivot Clustering has faster runtimes for larger amounts of streams. Upon checking, Pivot Clustering with estimated MSFEs for 200 streams takes approximately 20 min.

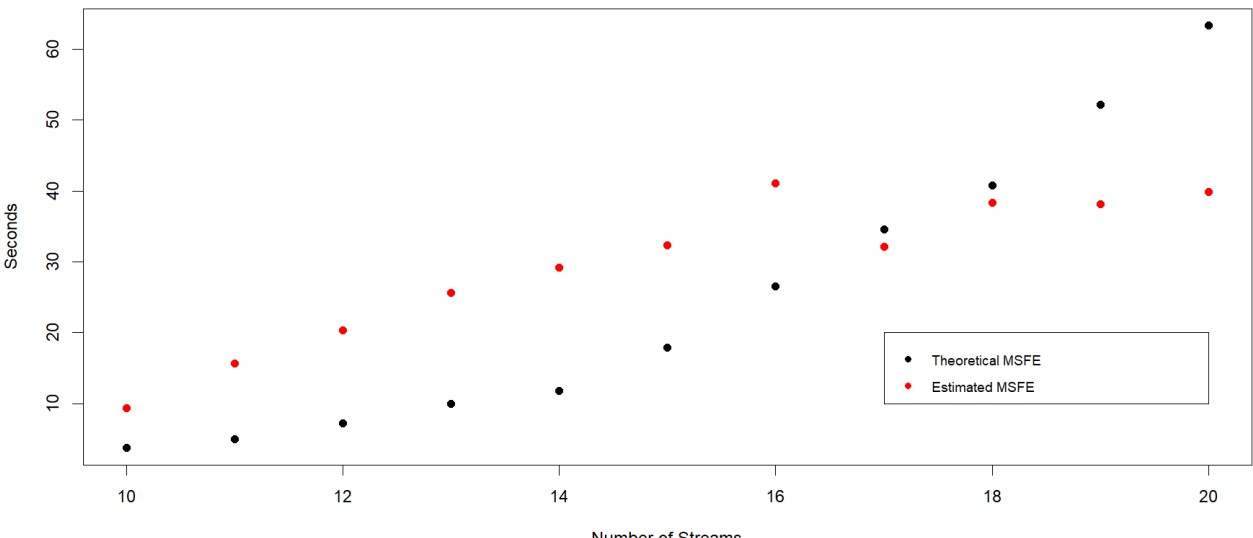

**Figure 1.** Runtimes of Pivot Clustering for assigning streams to four clusters using theoretical and estimated MSFEs. The graph contains the average runtime of Pivot Clustering for each specific number of streams being assigned to four clusters. That is, for *N* streams (between 10 and 20), we consider twenty randomly assigned ARMA(1,1) models for each stream. Pivot Clustering (with theoretical and estimated MSFE) is then used to obtain the four optimal clusters. The average runtime of Pivot Clustering is approximately 63 s when twenty streams are assigned to clusters using theoretical MSFE and 40 s using estimated MSFE.

We can check the efficacy of Pivot Clustering through simulation. We begin by randomly assigning coefficients to twenty ARMA(1,1) models to produce twenty demand streams as well as the covariance matrix of the shock sequences. We make sure that each assignment results in causal and invertible demand with respect to the shocks and that the resulting covariance matrix is positive definite. The AR and MA coefficients and covariance matrix can be found under "Models.csv" and "covarmat.csv" in our Github location [12].

After randomly assigning streams to one of four cluster,s we compute both the estimated and theoretical one-step-ahead MSFEs based on this random assignment and use it to start Pivot Clustering. We output the clusters found by Pivot Clustering as well as the MSFE of the forecast based on this set of clusters. We iterate this procedure 50 times to study how much the MSFE improves based on Pivot Clustering for the starting allocations. The MSFEs of the final clusters and random clusters can be found under "MSFEresults.csv" in our Github link [12]. These can also be compared with the MSFEs of the forecast based on the individual (disaggregated) demand streams and the forecast based on fully aggregating the streams.

For the twenty demand streams and models used, the theoretical and estimated MSFEs when forecasting based on individual (disaggregated) streams are 102.1 and 96.2. The theoretical and estimated MSFEs when forecasting based on the fully aggregated streams are 231.3 and 220.6. For the 50 simulations of assigning streams to random clusters (used in the initialization step of Pivot Clustering) the average of the theoretical and estimated MSFEs based on the subaggregated random clusters are 202.2 and 194.2. After Pivot Clustering is carried out to obtain a better set of subaggregated clusters in each of the 50 simulations, the averages of the theoretical and estimated MSFEs are 109.4 and 101.0. The various theoretical and estimated MSFEs for the different initializations are provided in Figures 2 and 3. We note that regardless of the initial random assignment of streams to clusters, Pivot Clustering leads to the clustering of streams such that the subaggregated MSFE is very low. In fact, typically, Pivot Clustering results in clusters for which the subaggregated MSFE ends up very close to the MSFE obtained when forecasts are based on the individual (disaggregated) streams.

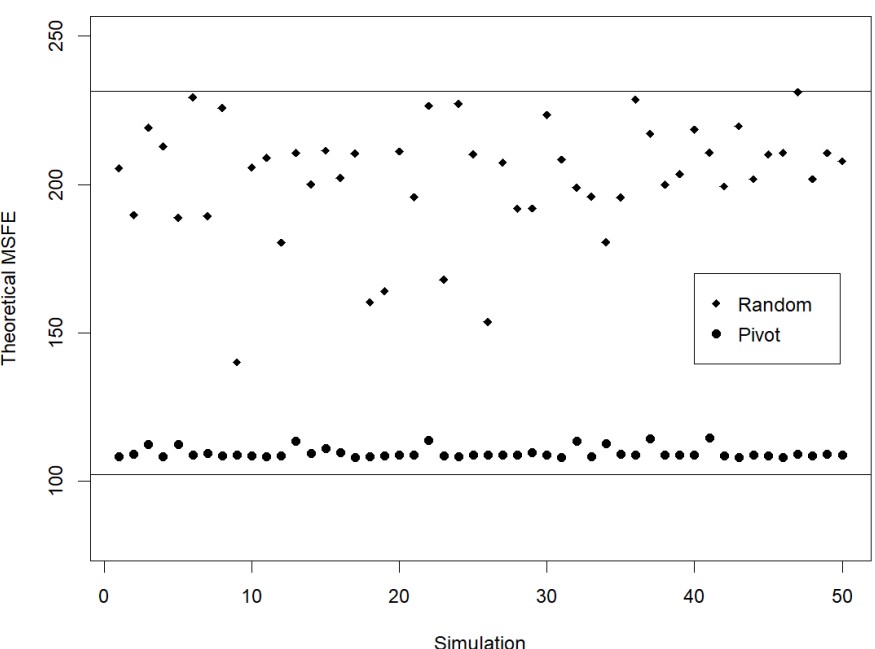

**Figure 2.** Theoretical subaggregated MSFE for random initialization of Pivot Clustering. Theoretical MSFEs are computed on the four clusters obtained by Pivot Clustering for different random initializations. The MSFE of the initial random assignment is provided as well as the MSFE that is obtained by Pivot Clustering. Horizontal lines are drawn to represent the MSFE based on the fully aggregated demand sequence (**top**) and the MSFE based on the fully disaggregated demand sequences (**bottom**).

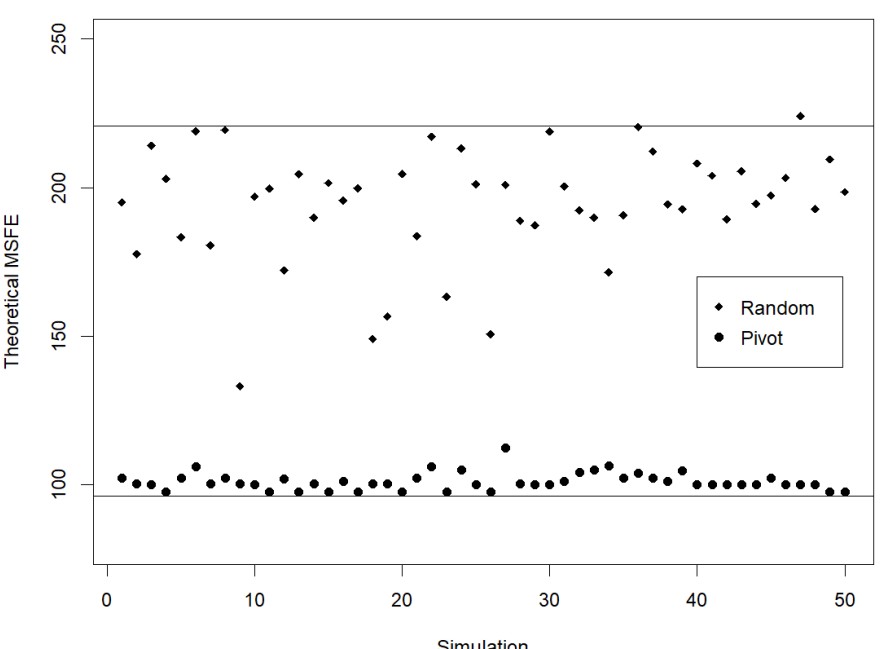

**Figure 3.** Estimated subaggregated MSFE for random initialization of Pivot Clustering. Estimated MSFEs are computed for different initializations of Pivot Clustering. The MSFE of the initial random assignment is provided as well as the MSFE that is obtained by Pivot Clustering. Horizontal lines are drawn to represent the MSFE based on the fully aggregated demand sequence (**top**) and the MSFE based on the fully disaggregated demand sequences (**bottom**).

We can compare our results with existing time-series clustering methods. Two distance measures that can be computed for time-series realizations are available in the TSclust package for R, namely AR.PIC and AR.LPC.CEPS. These distances can be used to perform hierarchical clustering such as average-linkage clustering. The final groups determined by these methods lead to MSFEs of 123.4 and 108.8, respectively, higher than those found by Pivot Clustering starting from random assignments. We note that the cluster assignments found by these methods can also be used in the initialization of Pivot Clustering, potentially leading to even better clusters.

Since the previous simulations were carried out on only one set of twenty ARMA(1,1) demand models, we should also check the efficacy of Pivot Clustering for other sets of models as well. As such, we consider twenty simulations where within each simulation a new set of twenty demand models is considered. We compare the estimated and theoretical MSFEs of one random assignment of streams to four clusters with the estimated and theoretical MSFEs of the four clusters obtained by Pivot Clustering. In each simulation, we also compute the MSFEs that would be found when fully aggregating the streams or when considering forecasts based on individual streams as well as the MSFEs that would be found using the AR.PIC and AR.LPC.CEPS distances for hierarchical clustering streams into four clusters. The results of these simulations are displayed in Figures 4 and 5. We note that if forecasts are to be based on four clusters, the lowest MSFEs are obtained when clusters are formed using Pivot Clustering. Furthermore, Pivot Clustering leads to forecasts whose MSFE is very close to the MSFE of the forecast based on the individual streams in every simulation.

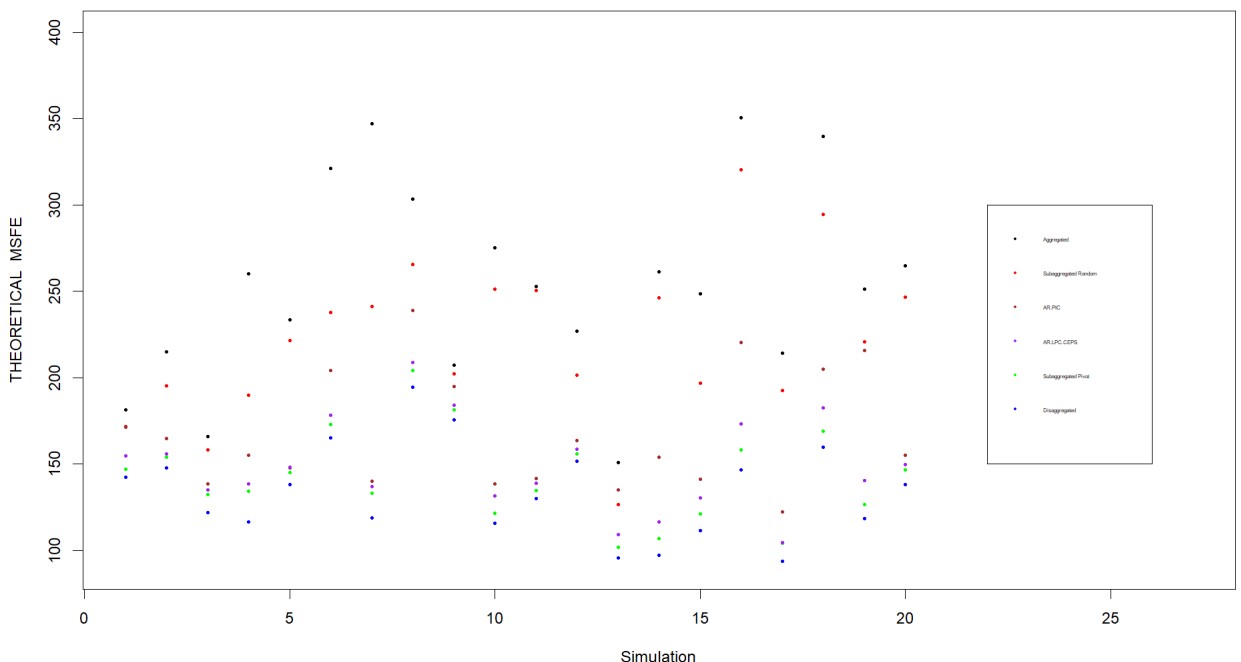

**Figure 4.** Theoretical subaggregated MSFE found by Pivot Clustering for different sets of streams. Theoretical MSFEs are computed for twenty simulations using different sets of twenty streams in each simulation. We note that using individual streams to forecasts leads to the lowest MSFE, while basing the forecast on the aggregate of the streams always leads to the highest MSFE. If subaggregated clusters are formed from the streams, the lowest MSFEs are obtained when clusters are based on Pivot Clustering.

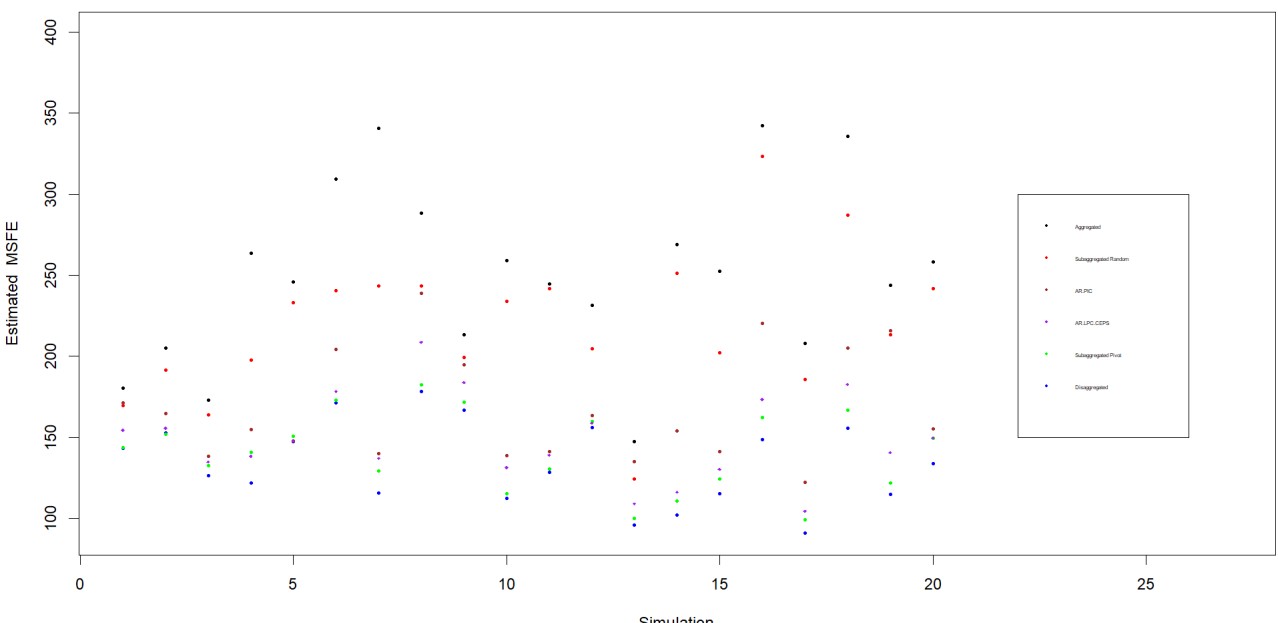

**Figure 5.** Estimated subaggregated MFSE found by Pivot Clustering for different sets of streams. Estimated MSFEs are computed for twenty simulations using different sets of twenty streams in each simulation. We note that using individual streams to forecasts leads to the lowest MSFE while basing the forecast on the aggregate of the streams always leads to the highest MSFE. If subaggregated clusters are formed from the streams, the lowest MSFEs are obtained when clusters are based on Pivot Clustering.

We continue with twenty simulations where in each simulation we consider a separate set of 20 streams being subaggregated into four clusters with 10 random initializations of Pivot Clustering. The means of the various theoretical and estimated MSFEs under different clustering approaches are displayed in Figures 6 and 7. We note again that in every set of twenty streams, the averaged subaggregated MSFEs are very close to the disaggregated MSFEs when averaged for different initial random assignments of streams to clusters.

We continue by assessing how well Pivot Clustering performs when compared against an exhaustive algorithm which checks all possible assignments of streams to clusters using theoretical MSFE calculations. To achieve this, we consider twenty simulations where within each simulation we randomly generate 10 ARMA(1,1) streams (We reduced the number of streams and clusters here due to the fact that an exhaustive algorithm requires $O(k^N)$ iterative steps to check all possible cluster assignments where $k$ is the number of clusters and $N$ is the number of streams. We note that Pivot Clustering has a complexity of $O(kN)$ in the event that each stream is only allowed to change clusters once.) and compute the lowest MSFE possible among all choices of streams to three clusters. Furthermore, we consider 10 random initializations of Pivot Clustering each time new streams are considered. The results are displayed in Table 2. For the first simulation of twenty streams (described by the first row of the table), 8 out of 10 initializations of our algorithm led Pivot Clustering to find the optimal solution. Among the two initializations that did not lead to the optimal solution, the ratio of optimal MSFE to the MSFE of the grouping found by pivot is 96.81%. The median was 96.81%, while the minimum ratio was 96.24%. In some instances Pivot Clustering never found an optimal solution (such as in the sixth simulation); however, the average MSFE of the optimal solution was around 99.6% of the MSFE of the groupings found by Pivot Clustering. In the worst performance, of Pivot Clustering (simulation 15), the best possible grouping led to an MSFE that was 80.27464% lower than the MSFE found by Pivot Clustering.

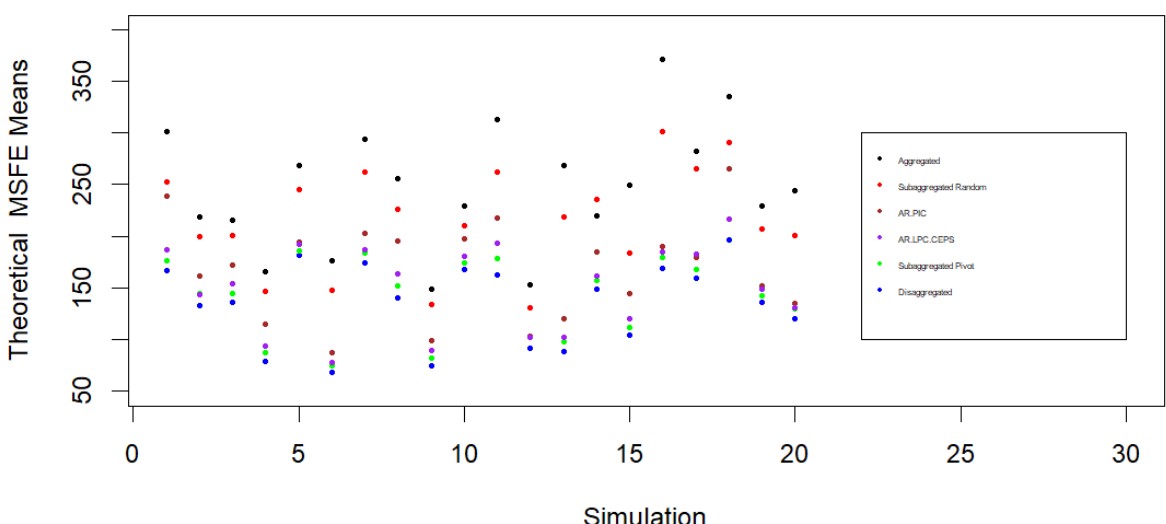

**Figure 6.** Theoretical subaggregated MSFE found by Pivot Clustering for different sets of streams. Theoretical MSFEs are computed for twenty simulations using different sets of twenty streams in each simulation and different initial assignments of streams to clusters. When averaging the final MSFEs based on the different initializations for each set of streams, we note that using individual streams to forecasts leads to the lowest averaged MSFE, while basing the forecast on the aggregate of the streams always leads to the highest averaged MSFE. If subaggregated clusters are formed from the streams, the lowest averaged MSFEs are obtained when clusters are based on Pivot Clustering.

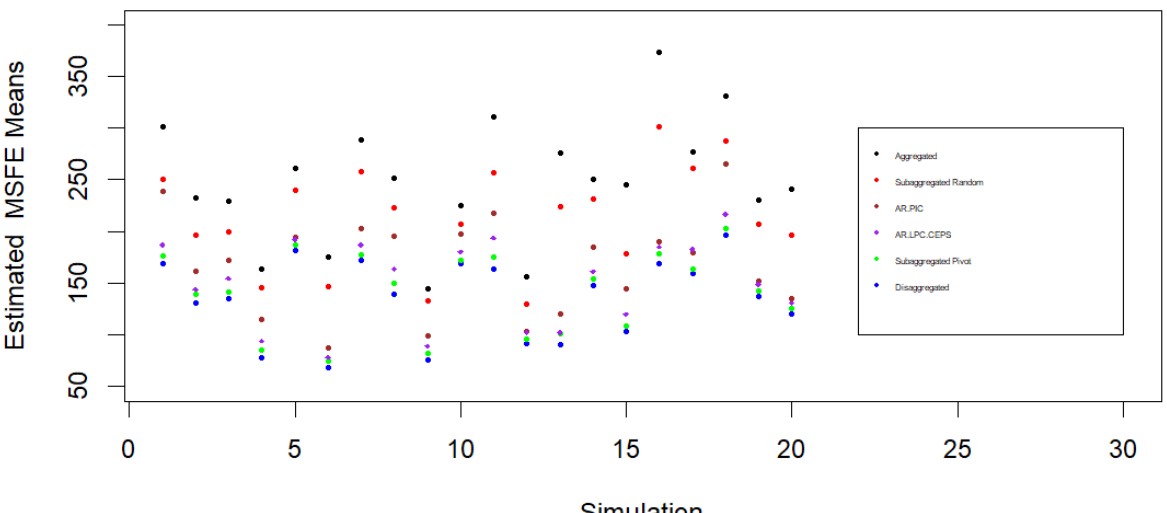

**Figure 7.** Estimated subaggregated MSFE found by Pivot Clustering for different sets of streams. Estimated MSFEs are computed for twenty simulations using different sets of twenty streams in each simulation and different initial assignments of streams to clusters. When averaging the final MSFEs based on the different initializations for each set of streams, we note that using individual streams to forecasts leads to the lowest averaged MSFE, while basing the forecast on the aggregate of the streams always leads to the highest averaged MSFE. If subaggregated clusters are formed from the streams, the lowest averaged MSFEs are obtained when clusters are based on Pivot Clustering.

**Table 2.** Twenty simulations are carried out where within each simulation we select a new set of ARMA(1,1) coefficients for each of 10 streams. The streams are then clustered using Pivot Clustering using theoretical MSFE based on 10 different starting groups. We also obtain the optimal (minimum MSFE) clustering assignment based on an exhaustive search of all possible assignments of streams to clusters. Each row corresponds to a new simulation. The three columns contain the mean, median and minimum ratios of global optimal MSFE to the MSFE obtained by Pivot Clustering for the different initializations in the event that Pivot Clustering did not find the optimal solution.

| Simulation | Average MSFE-Ratio of Global to Pivot | Median MSFE-Ratio of Global to Pivot | Minimum MSFE-Ratio of Global to Pivot |
|:---:|:---:|:---:|:---:|
| 1 | 0.9681 | 0.9681 | 0.9624 |
| 2 | 0.9774 | 0.9767 | 0.9748 |
| 3 | 0.9894 | 0.9890 | 0.9846 |
| 4 | 0.9865 | 0.9865 | 0.9865 |
| 5 | 0.9484 | 0.9479 | 0.9284 |
| 6 | 0.9959 | 0.9963 | 0.9952 |
| 7 | 0.9721 | 0.9810 | 0.9435 |
| 8 | 0.9933 | 0.9933 | 0.9933 |
| 9 | 0.9172 | 0.8920 | 0.8920 |
| 10 | 0.9809 | 0.9809 | 0.9809 |
| 11 | 0.9762 | 0.9861 | 0.9310 |
| 12 | 0.9902 | 0.9894 | 0.9864 |
| 13 | 0.9877 | 0.9877 | 0.9877 |
| 14 | 0.9890 | 0.9991 | 0.9688 |
| 15 | 0.8417 | 0.8481 | 0.8027 |
| 16 | 0.9568 | 0.9568 | 0.9315 |
| 17 | 0.9916 | 0.9952 | 0.9785 |
| 18 | 0.9942 | 0.9965 | 0.9802 |
| 19 | 0.9920 | 0.9948 | 0.9695 |
| 20 | 0.9946 | 0.9946 | 0.9946 |

Finally, we consider the robustness of the Pivot Clustering algorithm to cases where the data generation process is not ARMA. To accomplish this, we perform ten simulations where in each simulation we simulate twenty demand stream realizations such that stream $X_k$ follows an ARFIMA$(0, d_k, 0)$ model given by

$$(1 - B)^{d_k} X_{k,t} = \epsilon_{k,t} \tag{60}$$

where $-0.4 < d_k < 0.4$ and $Cov(\epsilon_{k,t}, \epsilon_{j,t})$ may be nonzero. Each realization, consisting of 1500 time periods, is used to fit an ARMA(5,5) model to compute an estimated one-step-ahead MSFE for the disaggregated series (appearing as a blue dot in Figure 8). Summing the realizations together to fit an ARMA(5,5) model yields an estimated MSFE for the aggregated series (appearing as a black dot in Figure 8). Finally, the Pivot Clustering algorithm is carried out using five different random initializations of assigning streams to one of four clusters. The MSFEs for the subaggregated random clusters and pivot clusters appear as red and green dots in Figure 8. We note that in the subaggregated case the number of ARMA models that needs to be estimated is equal to the number of clusters.

As before, we note that Pivot Clustering provides a sharp reduction in MSFE compared to random cluster assignments as well as compared to the aggregated case. We also observe that when fitting ARMA models to non-ARMA data, it is possible for Pivot Clustering to yield clusters which lead to a subaggregated MSFE that is lower than the MSFE using the individual (disaggregated) series. The exact cause of this is unclear; however, it is possible it has to do with the extra number of misspecified ARMA models that are fit in the disaggregated case.

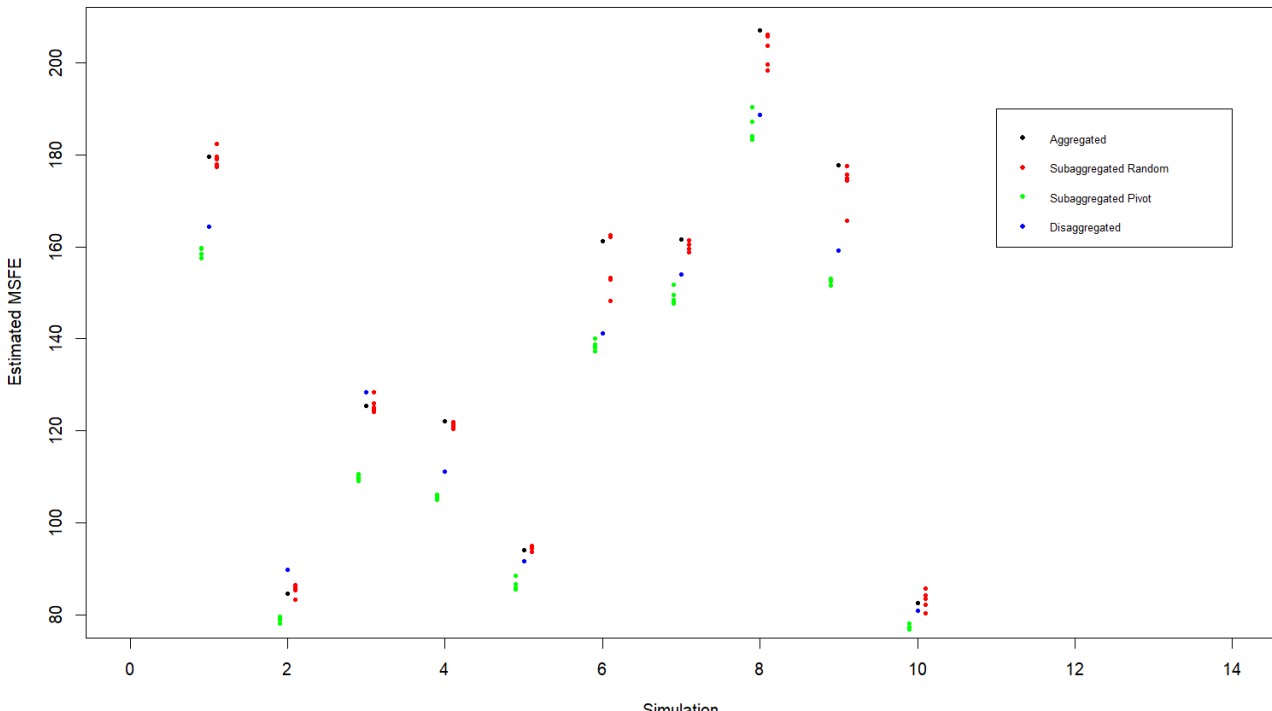

**Figure 8.** Estimated MSFEs when using aggregated, disaggregated and subaggregated series to forecast one-step-ahead demand for series that are not generated using an ARMA process. A total of 10 simulations are performed where, within each simulation, twenty separate demand realizations are generated according to the ARFIMA$(0, d, 0)$ model with a different $d$ for each realization such that the shocks appearing in the ARFIMA model are contemporaneously correlated. Pivot Clustering is carried out for five random initialization of assigning the streams to one of four clusters. We compute the estimated MSFEs for the disaggregated series (blue), aggregated series (black), subaggregated clusters generated using random assignment (red) and subaggregated clusters generated using the result of Pivot Clustering. We note the supremacy of Pivot Clustering in all ten simulations.

## 5. Clustering Demand Streams through Minimizing an Objective Function Based on Subaggregated MSFE

In this section, we describe how to determine the optimal assignment of streams to clusters by identifying and minimizing an objective function which computes the overall MSFE given a particular assignment of streams to clusters. We begin by assuming that the desired number of clusters is known to be $n$. For $\alpha \in 1, 2, \ldots n$, let subaggregated cluster series $\{C_{\alpha,t}\} = \{X_{1,t}y_{1,\alpha} + \ldots + X_{N,t}y_{N,\alpha}\}$, where $y_{i,\alpha} = 1$ if stream $\{X_{i,t}\}$ is in cluster $\{C_{\alpha,t}\}$ and 0 otherwise, have the ARMA representation

$$\phi_\alpha^\star(B)C_{\alpha,t} = \theta_\alpha^\star(B)\epsilon_{\alpha,t}^\star. \tag{61}$$

For $\alpha \in 1, 2, \ldots n$, the shocks $\epsilon_{\alpha,t}^\star$ have covariance matrix $\Sigma_\epsilon^\star$ given in Equation (42). We define the number of demand streams in cluster $\{C_{\alpha,t}\}$ to be $n_{C_\alpha} = \sum_{i=0}^{N} y_{i,\alpha} > 0$ for all $\alpha \in 1, 2, \ldots n$. Furthermore, if $y_{i,\alpha} = 1$ then $y_{i,\beta} = 0$ for all $\alpha \neq \beta \in 1, 2, \ldots n$. For each of $n_{C_\alpha}$ streams $\{X_{i,t}^{C_\alpha}\}$ in cluster $\{C_{\alpha,t}\}$, we adapt the notation of its ARMA representation to be

$$\Phi_{i,\alpha}(B)X_{i,t}^{C_\alpha} = \Theta_{i,\alpha}(B)\epsilon_{\alpha,i,t}. \tag{62}$$

**Lemma 1.** *An optimal set of clusters can be found by minimizing the subaggregated MSFE given by*

$$MSFEsubagg = \sum_{\alpha=1}^{n} \sum_{\beta=1}^{n} \sum_{l=0}^{\ell} \omega_{\beta,l}^{\star} \omega_{\alpha,l}^{\star} \cdot \sum_{i=1}^{n_{C_\alpha}} \sum_{j=1}^{n_{C_\beta}} \sum_{k=0}^{\infty} \tilde{\psi}_{\alpha,i,k} \tilde{\psi}_{\beta,j,k} \sigma_{ij} \tag{63}$$

*where $\tilde{\psi}_{\alpha,i,k}$ and $\tilde{\psi}_{\beta,j,k}$ are obtained from the equivalence*

$$\frac{\phi_\alpha^\star(z)}{\theta_\alpha^\star(z)} \frac{\Theta_{i,\alpha}(z)}{\Phi_{i,\alpha}(z)} \equiv \sum_{k=0}^{\infty} \tilde{\psi}_{\alpha,i,k} z^k \tag{64}$$

*and*

$$\frac{\phi_\beta^\star(z)}{\theta_\beta^\star(z)} \frac{\Theta_{j,\beta}(z)}{\Phi_{j,\beta}(z)} \equiv \sum_{k=0}^{\infty} \tilde{\psi}_{\beta,j,k} z^k. \tag{65}$$

*where the key terms are defined in the proof below.*

*Alternatively, the objective can be stated as finding the optimal set of $\{y_1, \dots, y_N\}$ such that we minimize*

$$MSFEsubagg = \sum_{\alpha=1}^{n} \sum_{\beta=1}^{n} \sum_{l=0}^{\ell} \omega_{\beta,l}^{\star} \omega_{\alpha,l}^{\star} \cdot \sum_{i=1}^{N} \sum_{j=1}^{N} \sum_{k=0}^{\infty} y_{i,\alpha} y_{j,\beta} \tilde{\psi}_{\alpha,i,k} \tilde{\psi}_{\beta,j,k} \sigma_{ij} \tag{66}$$

*where $\omega_{\alpha,l}^{\star}$ and $\psi_{\alpha,l}^{\star}$ in Equation (69) are obtained through the equivalence*

$$\frac{\theta_\alpha^\star(z)}{\phi_\alpha^\star(z)} = \sum_{l=0}^{\infty} \psi_{\alpha,l}^{\star} z^l \tag{67}$$

*where again the key terms are defined in subsequent proof.*

**Proof.** From Equation (7) we note that a forecast based on the clusters would have MSFE given by

$$MSFEsubagg = \sum_{\alpha=1}^{n} \sum_{\beta=1}^{n} \sum_{l=0}^{\ell} \sigma_{\alpha\beta}^{\star} \omega_{\beta,l}^{\star} \omega_{\alpha,l}^{\star} \tag{68}$$

where

$$\omega_{\alpha,l}^{\star} = \begin{cases} 0 & i < 0 \\ \psi_{\alpha,l}^{\star} & l = 0 \\ \omega_{\alpha,l-1}^{\star} + \psi_{\alpha,l}^{\star} & 0 < l < \ell+1 \\ \omega_{\alpha,l-1}^{\star} + \psi_{\alpha,l}^{\star} - \psi_{\alpha,l-\ell-1}^{\star} & l \geq \ell+1. \end{cases} \tag{69}$$

where $\psi_{\alpha,l}^{\star}$ is the $l$th coefficient appearing in the MA($\infty$) representation of $\{C_{\alpha,t}\}$ with respect to $\{\epsilon_{\alpha,t}^{\star}\}$. From Equation (41), we note that for any two subaggregated clusters $\{C_{\alpha,t}\}$ and $\{C_{\beta,t}\}$ consisting of $n_{C_\alpha}$ and $n_{C_\beta}$ streams, respectively, the corresponding shock series $\{\epsilon_{\alpha,t}^{\star}\}$ and $\{\epsilon_{\beta,t}^{\star}\}$, the covariance $E[\epsilon_{\alpha,t}^{\star} \epsilon_{\beta,t}^{\star}]$ is expressed by

$$\sigma_{\alpha\beta}^{\star} = \sum_{i=1}^{n_{C_\alpha}} \sum_{j=1}^{n_{C_\beta}} \sum_{k=0}^{\infty} \tilde{\psi}_{\alpha,i,k} \tilde{\psi}_{\beta,j,k} \sigma_{ij}. \tag{70}$$

Therefore, the objective is to assign streams to clusters such that we minimize the MSFE

$$MSFEsubagg = \sum_{\alpha=1}^{n} \sum_{\beta=1}^{n} \sum_{l=0}^{\ell} \omega_{\beta,l}^{\star} \omega_{\alpha,l}^{\star} \cdot \sum_{i=1}^{n_{C_\alpha}} \sum_{j=1}^{n_{C_\beta}} \sum_{k=0}^{\infty} \tilde{\psi}_{\alpha,i,k} \tilde{\psi}_{\beta,j,k} \sigma_{ij} \tag{71}$$

where $\tilde{\psi}_{\alpha,i,k}$ and $\tilde{\psi}_{\beta,j,k}$ are obtained from the equivalence

$$\frac{\phi_\alpha^\star(z)}{\theta_\alpha^\star(z)} \frac{\Theta_{i,\alpha}(z)}{\Phi_{i,\alpha}(z)} \equiv \sum_{k=0}^\infty \tilde{\psi}_{\alpha,i,k} z^k \tag{72}$$

and

$$\frac{\phi_\beta^\star(z)}{\theta_\beta^\star(z)} \frac{\Theta_{j,\beta}(z)}{\Phi_{j,\beta}(z)} \equiv \sum_{k=0}^\infty \tilde{\psi}_{\beta,j,k} z^k. \tag{73}$$

Alternatively, we can say that we are finding the optimal set of $\{y_1, \ldots, y_N\}$ such that we minimize

$$MSFEsubagg = \sum_{\alpha=1}^n \sum_{\beta=1}^n \sum_{l=0}^\ell \omega_{\beta,l}^\star \omega_{\alpha,l}^\star \cdot \sum_{i=1}^N \sum_{j=1}^N \sum_{k=0}^\infty y_{i,\alpha} y_{j,\beta} \tilde{\psi}_{\alpha,i,k} \tilde{\psi}_{\beta,j,k} \sigma_{ij} \tag{74}$$

where $\omega_{\alpha,l}^\star$ and $\psi_{\alpha,l}^\star$ in Equation (69) are obtained through the equivalence

$$\frac{\theta_\alpha^\star(z)}{\phi_\alpha^\star(z)} = \sum_{l=0}^\infty \psi_{\alpha,l}^\star z^l \tag{75}$$

with $\theta_\alpha^\star(z)$ and $\phi_\alpha^\star(z)$ found using Remark 1, where Laurent polynomials $O(z), P(z)$ and $Q(z)$ are obtained from the covariance generating function $G_{C_\alpha}(z)$ given by

$$\begin{aligned} G_{C_\alpha}(z) &= \sum_{i=1}^N y_{i,\alpha} \sigma_i^2 \frac{\Theta_i(z)\Theta_i(z^{-1})}{\Phi_i(z)\Phi_i(z^{-1})} \\ &+ \sum_{i=1}^{N-1} \sum_{j=i+1}^N y_{i,\alpha} y_{j,\alpha} \left( \sigma_{ij} \frac{\Theta_i(z)}{\Phi_i(z)} \frac{\Theta_j(z^{-1})}{\Phi_j(z^{-1})} + \sigma_{ij} \frac{\Theta_i(z^{-1})}{\Phi_i(z^{-1})} \frac{\Theta_j(z)}{\Phi_j(z)} \right). \end{aligned} \tag{76}$$

□

We note that it is impossible to offer an explicit solution because of the dependence of coefficients $\omega_{\beta,l}^\star, \omega_{\alpha,l}^\star, \tilde{\psi}_{\alpha,i,k}$ and $\tilde{\psi}_{\beta,j,k}$ on the selection of clusters. In the next section, we consider a much simpler case of demand streams being generated by MA(1) models which leads to a much simpler objective function. This allows us to find several theoretical results, culminating in the fact that optimal clusters can be found in this case by identifying streams having the closest MA coefficients with one another.

## 6. MA(1) Streams

In this section, we consider the case that the demand streams being considered are independent MA(1). As we demonstrate below, this leads to a simpler objective function. We will use this fact to show how we can use non-linear optimization to assign clusters to streams and to come up with an efficient way to cluster independent MA(1) streams based on segmenting the coefficient space into intervals. The focus on MA(1) streams here allows us to observe that streams with their MA coefficients close to each other should be clustered together. At the end of the section, we provide a lemma on aggregating streams produced by models having identical ARMA coefficients.

**Lemma 2.** *Suppose* $\{X_{1,t}\}, \{X_{2,t}\}, \ldots, \{X_{N,t}\}$ *are MA(1) with MA coefficients* $\theta_1, \theta_2, \ldots, \theta_N$. *Optimal clusters can be found by assigning* $y_{jk}$ *as an indicator variable for stream* $X_j$ *being in cluster* $C_k$ *such that we minimize*

$$\sum_{k=1}^n \sqrt{(b_k + 2a_k)(b_k - 2a_k)} \tag{77}$$

*where*

$$b_k = \sum_{j=1}^{N} \sigma_j^2 (1 + \theta_j^2) y_{jk} \qquad (78)$$

*and*

$$a_k = \sum_{j=1}^{N} \sigma_j^2 \theta_j y_{jk}. \qquad (79)$$

*Alternatively, the objective function (77) can be written as*

$$\sum_{k=1}^{n} \sqrt{\sum_{j=1}^{N} \sum_{i=1}^{N} \sigma_j^2 \sigma_i^2 (1 + \theta_j)^2 (1 - \theta_i)^2 y_{jk} y_{ik}}. \qquad (80)$$

**Proof.** Suppose $\{X_{1,t}\}, \{X_{2,t}\}, \ldots, \{X_{N,t}\}$ are MA(1) with MA coefficients $\theta_1, \theta_2, \ldots, \theta_N$. Suppose cluster $C_{\alpha,t}$ consists of streams $\{X_{1,t}\}, \ldots, \{X_{\alpha,t}\}$. The covariance generating function of $C_{\alpha,t}$ simplifies to

$$G_{C_\alpha}(z) = \sigma_1^2 (1 + \theta_1 z)(1 + \theta_1 z^{-1}) + \ldots + \sigma_\alpha^2 (1 + \theta_\alpha z)(1 + \theta_\alpha z^{-1}). \qquad (81)$$

In order to determine the variance of the shocks of $C_{\alpha,t}$, we need to find the roots of Equation (81). Note that it can be rewritten as

$$\left( \sigma_1^2 \theta_1 + \ldots + \sigma_\alpha^2 \theta_\alpha \right) z^{-1} + \left( \sigma_1^2 (1 + \theta_1^2) + \ldots + \sigma_\alpha^2 (1 + \theta_\alpha^2) \right) + \left( \sigma_1^2 \theta_1 + \ldots + \sigma_\alpha^2 \theta_\alpha \right) z. \quad (82)$$

We can find the roots of Equation (82) using the quadratic formula $\dfrac{-b \pm \sqrt{b^2 - 4a^2}}{2a}$ where

$$a \;=\; \sigma_1^2 \theta_1 + \ldots + \sigma_\alpha^2 \theta_\alpha = \sum_{j=1}^{\alpha} \sigma_j^2 \theta_j \qquad (83)$$

$$b \;=\; \sigma_1^2 (1 + \theta_1^2) + \ldots + \sigma_\alpha^2 (1 + \theta_\alpha^2) = \sum_{j=1}^{\alpha} \sigma_j^2 (1 + \theta_j^2) \qquad (84)$$

We note that one roots $r_1$ will be outside the unit circle and from Remark 1 that the variance of the shocks of $C_{\alpha,t}$ is equal to $-ar_1$. Since $b > 0$, the variance is then given by

$$\sigma_\alpha^2 = \frac{b + \sqrt{(b + 2a)(b - 2a)}}{2} \qquad (85)$$

Thus, if we are subaggregating into $n$ clusters, the MSFE is given by

$$MSFE_{subagg} = \frac{1}{2} \left( \sum_{k=1}^{n} b_k + \sum_{k=1}^{n} \sqrt{(b_k + 2a_k)(b_k - 2a_k)} \right) \qquad (86)$$

Since $\displaystyle\sum_{k=1}^{n} b_k$ will be the same regardless of choice of clusters, we are minimizing the objective function given by

$$\sum_{k=1}^{n} \sqrt{(b_k + 2a_k)(b_k - 2a_k)} \qquad (87)$$

*where*

$$b_k = \sum_{j=1}^{N} \sigma_j^2 (1 + \theta_j^2) y_{jk} \qquad (88)$$

and

$$a_k = \sum_{j=1}^{N} \sigma_j^2 \theta_j y_{jk} \tag{89}$$

where $y_{jk}$ is an indicator variable for stream $X_j$ being in cluster $C_k$.

Noting that

$$(b_k + 2a_k) = \sum_{j=1}^{N} \sigma_j^2 (1 + \theta_j)^2 y_{jk} \tag{90}$$

$$(b_k - 2a_k) = \sum_{j=1}^{N} \sigma_j^2 (1 - \theta_j)^2 y_{jk} \tag{91}$$

The objective function can also be rewritten as

$$\sum_{k=1}^{n} \sqrt{\sum_{j=1}^{N} \sum_{i=1}^{N} \sigma_j^2 \sigma_i^2 (1 + \theta_j)^2 (1 - \theta_i)^2 y_{jk} y_{ik}}. \tag{92}$$

$\square$

We have results (not shown here) where we use Equation (92) in a non-linear optimization algorithm to come up with clusters. When using 10 streams with three clusters, the approach actually leads to a globally optimal solution almost every time.

We note that Equation (92) can also be rewritten as

$$\sum_{k=1}^{n} \sqrt{\sum_{X_i, X_j \in C_k} \sigma_j^2 \sigma_i^2 (1 + \theta_j)^2 (1 - \theta_i)^2}. \tag{93}$$

where the inner sum is taken over all pairs of streams in each cluster, including pairs of streams with themselves.

If stream $X_p$ is moved from cluster $k$ to cluster $\kappa$, then the change in the objective function is

$$\sqrt{\sigma_p^2 (1 + \theta_p)^2 \sum_{X_i \in C_\kappa} \sigma_i^2 (1 - \theta_i)^2 + \sigma_p^2 (1 - \theta_p)^2 \sum_{X_i \in C_\kappa} \sigma_i^2 (1 + \theta_i)^2}$$
$$- \sqrt{\sigma_p^2 (1 + \theta_p)^2 \sum_{X_i \in C_k} \sigma_i^2 (1 - \theta_i)^2 + \sigma_p^2 (1 - \theta_p)^2 \sum_{X_i \in C_k} \sigma_i^2 (1 + \theta_i)^2} \tag{94}$$

which provides an alternative way to cluster streams by identifying and moving the stream from one cluster to another, which yields the largest drop in the objective function.

*Aggregate of Two MA(1) Streams*

In this subsection, we consider two MA(1) streams whose variance of shocks is unitary. We demonstrate that the MA coefficient (The aggregate of two MA(1) streams is always MA(1)) of their aggregate process is always between the two MA coefficients of the individual streams. Furthermore, we show that as the two coefficients of the two MA(1) streams are moved further apart from each other, the variance of the shocks appearing in the aggregated process increases. These two facts imply that if we are studying $N$ individual MA(1) streams (with unit shock variance) and would like to cluster them into $n$ clusters, then the globally optimal clustering assignment will cluster the streams along intervals. That is, the assignment will have split the clusters into groups of streams whose MA coefficients are next to each other in a sorted arrangement. This implies that an efficient algorithm for obtaining a globally optimal cluster assignment consists of arranging the MA coefficients in increasing order and checking all possible "interval" clusters, without worrying that if

two streams are clustered together, another stream with an MA coefficient between the two is assigned to a different cluster. This algorithm would need to check $\binom{N}{n}$ possible cluster assignments to find the globally optimal arrangement.

**Theorem 1.** *Consider two streams whose ARMA representations are given by*

$$X_{1,t} = (1 + \theta_1 B)\epsilon_{1,t}$$

$$X_{2,t} = (1 + \theta_2 B)\epsilon_{2,t}$$

*with* $var(\epsilon_{1,t}) = var(\epsilon_{2,t}) = 1$ *and* $\sigma_{12} = 0$ *and* $\theta_1 < \theta_2$. *Note that* $\theta_1$ *and* $\theta_2$ *are allowed to equal 0.*

*The aggregated process* $\{X_{1,t} + X_{2,t}\}$ *is described by the MA(1) model*

$$X_{1,t} + X_{2,t} = (1 + \theta B)\epsilon_t$$

*such that* $\theta_1 < \theta < \theta_2$.

**Proof.** Note that the covariance generating function of the aggregate process is

$$G_S(z) = (\theta_1 + \theta_2)z^{-1} + (2 + \theta_1^2 + \theta_2^2) + (\theta_1 + \theta_2)z. \tag{95}$$

As long as $\theta_1 \neq -\theta_2$ (if $\theta_1 = -\theta_2$, then $\theta = 0$ and the result still holds), this polynomial has roots $a_1$ and $1/a_1$. Suppose $a_1$ is inside the unit circle, then according to Remark 1, $\theta = -a_1$ and $var(\epsilon_t) = \sigma_\epsilon^2 = -(\theta_1 + \theta_2)(1/a_1)$.

We can note that the Laurent polynomial in Equation (95) has roots given by

$$\frac{-(2 + \theta_1^2 + \theta_2^2) \pm \sqrt{(2 + \theta_1^2 + \theta_2^2)^2 - 4(\theta_1 + \theta_2)^2}}{2(\theta_1 + \theta_2)} \tag{96}$$

This implies that

$$a_1 = \frac{-(2 + \theta_1^2 + \theta_2^2) + \sqrt{(2 + \theta_1^2 + \theta_2^2)^2 - 4(\theta_1 + \theta_2)^2}}{2(\theta_1 + \theta_2)} \tag{97}$$

and

$$\theta = \frac{(2 + \theta_1^2 + \theta_2^2) - \sqrt{(2 + \theta_1^2 + \theta_2^2)^2 - 4(\theta_1 + \theta_2)^2}}{2(\theta_1 + \theta_2)} \tag{98}$$

In the remainder of this section, we will prove that

$$\theta_1 < \frac{(2 + \theta_1^2 + \theta_2^2) - \sqrt{(2 + \theta_1^2 + \theta_2^2)^2 - 4(\theta_1 + \theta_2)^2}}{2(\theta_1 + \theta_2)} < \theta_2 \tag{99}$$

Suppose first that $\theta_1 + \theta_2 > 0$.
Then, we can rewrite Equation (99) as

$$2\theta_1^2 + 2\theta_2\theta_1 < (2 + \theta_1^2 + \theta_2^2) - \sqrt{(2 + \theta_1^2 + \theta_2^2)^2 - 4(\theta_1 + \theta_2)^2} < 2\theta_2^2 + 2\theta_2\theta_1 \tag{100}$$

or equivalently,

$$\theta_1^2 - \theta_2^2 + 2\theta_2\theta_1 - 2 < -\sqrt{(2 + \theta_1^2 + \theta_2^2)^2 - 4(\theta_1 + \theta_2)^2} < \theta_2^2 - \theta_1^2 + 2\theta_1\theta_2 - 2 \tag{101}$$

or

$$\theta_1^2 - \theta_2^2 - 2\theta_2\theta_1 + 2 < \sqrt{(2 + \theta_1^2 + \theta_2^2)^2 - 4(\theta_1 + \theta_2)^2} < \theta_2^2 - \theta_1^2 - 2\theta_1\theta_2 + 2 \tag{102}$$

Noting that the left and right-hand sides of the inequality are always larger than zero, (102) holds if and only if the following inequality holds as well:

$$(\theta_1^2 - \theta_2^2 - 2\theta_2\theta_1 + 2)^2 < (2 + \theta_1^2 + \theta_2^2)^2 - 4(\theta_1 + \theta_2)^2 < (\theta_2^2 - \theta_1^2 - 2\theta_1\theta_2 + 2)^2. \quad (103)$$

Labeling the three sides of this inequality as $A < B < C$, we observe that

$$
\begin{aligned}
A \;=\; & \theta_1^4 - \theta_1^2\theta_2^2 - 2\theta_1^3\theta_2 + 2\theta_1^2 - \theta_1^2\theta_2^2 + \theta_2^4 + 2\theta_2^3\theta_1 - 2\theta_2^2 \\
& - \; 2\theta_1^3\theta_2 + 2\theta_2^3\theta_1 + 4\theta_1^2\theta_2^2 - 4\theta_2\theta_1 + 2\theta_1^2 - 2\theta_2^2 - 4\theta_2\theta_1 + 4 \\
B \;=\; & 4 + 2\theta_1^2 + 2\theta_2^2 + 2\theta_1^2 + \theta_1^4 + \theta_1^2\theta_2^2 + 2\theta_2^2 + \theta_1^2\theta_2^2 + \theta_2^4 - 4(\theta_1^2 + 2\theta_1\theta_2 + \theta_2^2) \\
C \;=\; & \theta_2^4 - \theta_2^2\theta_1^2 - 2\theta_2^3\theta_1 + 2\theta_2^2 - \theta_2^2\theta_1^2 + \theta_1^4 + 2\theta_1^3\theta_2 - 2\theta_1^2 \\
& - \; 2\theta_2^3\theta_1 + 2\theta_1^3\theta_2 + 4\theta_2^2\theta_1^2 - 4\theta_1\theta_2 + 2\theta_2^2 - 2\theta_1^2 - 4\theta_1\theta_2 + 4
\end{aligned}
\quad (104)
$$

Removing equivalent terms and combining like terms, we observe

$$
\begin{aligned}
& 2\theta_1^2\theta_2^2 - 4\theta_1^3\theta_2 + 4\theta_1^2 + 4\theta_2^3\theta_1 - 4\theta_2^2 - 8\theta_1\theta_2 \\
< \;& 2\theta_1^2\theta_2^2 - 8\theta_1\theta_2 \\
< \;& \theta_2^2\theta_1^2 - 4\theta_2^3\theta_1 + 4\theta_2^2 + 4\theta_1^3\theta_2 - 4\theta_1^2 - 8\theta_2\theta_1
\end{aligned}
$$

which can be rewritten as

$$-4\theta_1^3\theta_2 + 4\theta_1^2 + 4\theta_1\theta_2^3 - 4\theta_2^2 < 0 < -4\theta_1\theta_2^3 + 4\theta_2^2 + 4\theta_1^3\theta_2 - 4\theta_1^2. \quad (105)$$

Noting that the left and right-hand sides of this inequality are additive inverses, we see that this inequality holds (the given direction of the inequalities must hold, otherwise $\theta_1 > \theta_2$ in (99)) and therefore inequality (99) holds.

Finally, if $\theta_1 + \theta_2 < 0$ in (99), then the direction of the inequalities is reversed in (100) and a similar sequence of steps would lead us to observe

$$-4\theta_1^3\theta_2 + 4\theta_1^2 + 4\theta_1\theta_2^3 - 4\theta_2^2 > 0 > -4\theta_1\theta_2^3 + 4\theta_2^2 + 4\theta_1^3\theta_2 - 4\theta_1^2 \quad (106)$$

and by the same argument, we see that (99) holds and the theorem is proved. $\quad\square$

**Theorem 2.** *Consider two streams whose ARMA representations are given by*

$$X_{1,t} = (1 + \theta_1 B)\epsilon_{1,t}$$

$$X_{2,t} = (1 + \theta_2 B)\epsilon_{2,t}$$

*with $\sigma_1^2 = \sigma_2^2 = 1$ and $\sigma_{12} = 0$ and $\theta_1 < \theta_2$. Note that $\theta_1$ and $\theta_2$ may be 0.*
*As the distance between $\theta_1$ and $\theta_2$ increases, $var(\epsilon_t) = \sigma_\epsilon^2$ increases.*

**Proof.** From Equation (95) we see that the root $1/a_1$ of $G_S(z)$, which is outside the unit circle, is given by

$$\frac{-(2 + \theta_1^2 + \theta_2^2) - \sqrt{(2 + \theta_1^2 + \theta_2^2)^2 - 4(\theta_1 + \theta_2)^2}}{2(\theta_1 + \theta_2)}. \quad (107)$$

Since $\sigma_\epsilon^2 = -(\theta_1 + \theta_2)/a_1$, we have

$$\sigma_\epsilon^2 = \frac{(2 + \theta_1^2 + \theta_2^2) + \sqrt{(2 + \theta_1^2 + \theta_2^2)^2 - 4(\theta_1 + \theta_2)^2}}{2} \quad (108)$$

To show that this is increasing in $\theta_2$, consider the derivative of the above with respect to $\theta_2$:

$$\frac{\partial \sigma_\epsilon^2}{\partial \theta_2} = \theta_2 + \frac{\theta_2(2 + \theta_1^2 + \theta_2^2) - 2(\theta_1 + \theta_2)}{\sqrt{(2 + \theta_1^2 + \theta_2^2)^2 - 4(\theta_1 + \theta_2)^2}}. \tag{109}$$

We need to show that (109) is larger than zero, thus consider

$$\frac{\partial \sigma_\epsilon^2}{\partial \theta_2} = \theta_2 + \frac{\theta_2(2 + \theta_1^2 + \theta_2^2) - 2(\theta_1 + \theta_2)}{\sqrt{(2 + \theta_1^2 + \theta_2^2)^2 - 4(\theta_1 + \theta_2)^2}} > 0 \tag{110}$$

or equivalently

$$\theta_2(2 + \theta_1^2 + \theta_2^2) - 2(\theta_1 + \theta_2) > -\theta_2\sqrt{(2 + \theta_1^2 + \theta_2^2)^2 - 4(\theta_1 + \theta_2)^2} \tag{111}$$

which simplifies to

$$2(\theta_1 + \theta_2) - \theta_2(2 + \theta_1^2 + \theta_2^2) < \theta_2\sqrt{(2 + \theta_1^2 + \theta_2^2)^2 - 4(\theta_1 + \theta_2)^2} \tag{112}$$

or

$$2\theta_1 - \theta_2\theta_1^2 - \theta_2^3 < \theta_2\sqrt{(2 + \theta_1^2 + \theta_2^2)^2 - 4(\theta_1 + \theta_2)^2}. \tag{113}$$

We will refer to the left and right hand sides of the inequality in (113) as $LHS$ and $RHS$. Note that squaring both yields,

$$\begin{aligned} LHS^2 &= 4\theta_1^2 - 4\theta_1^3\theta_2 - 4\theta_1\theta_2^3 + \theta_1^4\theta_2^2 + 2\theta_1^2\theta_2^4 + \theta_2^6 \\ RHS^2 &= 4\theta_2^2 + 4\theta_1^2\theta 2^2 + \theta_1^4\theta_2^2 + 2\theta_1^2\theta_2^4 + \theta_2^6 - 4\theta_1^2\theta_2^2 - 8\theta_1\theta_2^3 \end{aligned} \tag{114}$$

Note that, furthermore,

$$LHS^2 - RHS^2 = 4\theta_1^2 - 4\theta_2^2 - 4\theta_1^3\theta_2 + 4\theta_1\theta_2^3 = 4(1 - \theta_1\theta_2)(\theta_1^2 - \theta_2^2) \tag{115}$$

Suppose first that $\theta_1 < \theta_2 = 0$. Note that (113) reduces to $2\theta_1 < 0$, which holds, and therefore (110) holds as well.

Next, suppose that $0 = \theta_1 < \theta_2$. Note that (113) reduces to

$$-\theta_2^3 < \theta_2\sqrt{(2 + \theta_2^2)^2 - 4\theta_2^2} \tag{116}$$

which holds as well since the left-hand side is negative in this case.

In the remainder of the proof, we assume that $\theta_1 \neq 0$ and $\theta_2 \neq 0$. Consider the case that $|\theta_2| > |\theta_1|$. Note that this implies that $\theta_2 > 0$ (since $\theta_2 > \theta_1$) and that (115) is less than zero. Thus, in this case $LHS^2 < RHS^2$ and $LHS < RHS$ in (113). Therefore, (110) holds in this case as well.

Now, suppose that $|\theta_1| > |\theta_2|$ and that $\theta_2 > 0$. The former implies that $LHS^2 - RHS^2 > 0$ and the latter implies that $RHS > 0$. Therefore $LHS < 0$ and therefore $LHS < RHS$. Therefore, (110) holds in this case as well.

Finally, suppose that $|\theta_1| > |\theta_2|$ and that $\theta_2 < 0$. The former again implies that $LHS^2 - RHS^2 > 0$, while the latter implies that $RHS < 0$. Therefore, $LHS < RHS$ and (110) holds in all cases. $\square$

## 7. Demand Streams Produced by Identical ARMA Models

Note that Theorems 1 and 2 imply that the prescribed algorithm at the top of the previous subsection always leads to an optimal solution. The following lemma establishes that in the event that two streams are generated by the same ARMA model, the aggregate will also follow the same ARMA model. Therefore, we can greatly reduce the dimensionality

of the number of streams that need to be assigned to clusters by first aggregating demand streams from equivalent models.

**Lemma 3.** *Consider two sequences* $\{X_{1,t}\}$ *and* $\{X_{2,t}\}$ *that have the same ARMA representation with respect to Wold shock sequences* $\{\epsilon_{1,t}\}$ *and* $\{\epsilon_{2,t}\}$ *given by*

$$\Phi^\star(B)X_{1,t} = \Theta^\star(B)\epsilon_{1,t} \tag{117}$$
$$\Phi^\star(B)X_{2,t} = \Theta^\star(B)\epsilon_{2,t} \tag{118}$$

*such that the variance of the shock sequences are* $\sigma_1^2$ *and* $\sigma_2^2$ *with covariance* $\sigma_{12}$.

*The aggregate* $\{S_t = X_{1,t} + X_{2,t}\}$ *also has the same ARMA representation with respect to its Wold shocks* $\{\epsilon_t\}$ *given by*

$$\Phi^\star(B)S_t = \Theta^\star(B)\epsilon_t \tag{119}$$

*such that the variance of* $\{\epsilon_t\}$ *is given by* $\sigma_1^2 + \sigma_2^2 + 2\sigma_{12}$.

**Proof.** From Remark 1, we note that the ARMA representation of $\{S_t\}$ is given by

$$\Phi(B)S_t = \Theta(B)\epsilon_t \tag{120}$$

such that $\Theta(z) = \prod_{i=1}^{m}(1 - a_i z)$ where $\{a_i\}$ are the roots of $O(z)P(z)$ on or inside the unit circle and $\Phi(z) = \prod_{i=1}^{n}(1 - b_i z)$ where $\{b_i\}$ are the roots of $Q(z)$ inside the unit circle with $O(z), P(z)$ and $Q(z)$ are obtained from the covariance generating function $G_S(z)$ given by

$$G_S(z) = (\sigma_1^2 + \sigma_2^2)\frac{\Theta^\star(z)\Theta^\star(z^{-1})}{\Phi^\star(z)\Phi^\star(z^{-1})} + 2\sigma_{12}\frac{\Theta^\star(z)\Theta^\star(z^{-1})}{\Phi^\star(z)\Phi^\star(z^{-1})} \tag{121}$$

as per (76). This can further be simplified as

$$G_S(z) = (\sigma_1^2 + \sigma_2^2 + 2\sigma_{12})\frac{\Theta^\star(z)\Theta^\star(z^{-1})}{\Phi^\star(z)\Phi^\star(z^{-1})} \tag{122}$$

and therefore $\Phi(z) = \Phi^\star(z)$ and $\Theta(z) = \Theta^\star(z)$, and the result is proved. □

Lemma 3 shows us that if we have $n$ demand sequences $X_{1,t}, \ldots, X_{n,t}$, generated by models with the same ARMA coefficients with respect to their Wold shocks, their aggregate will have the same ARMA coefficients. Therefore, if the customer base of a firm is comprised of many demand streams having the same ARMA representation, it is possible to greatly reduce the number of streams that need to be considered for clustering by first aggregating these equivalent streams.

## 8. Extensions and Other Questions

In this paper we compare theoretical MSFEs of a firm forecasting its leadtime demand based on disaggregated (individual) demand streams and subaggregated clusters formed from those streams. We highlight examples that illustrate that the MSFE based on subaggregates need not be much larger than the MSFE based on the disaggregated streams as long as those clusters are well formed. We propose a Pivot Clustering algorithm to form clusters which minimize the MSFE among all cluster assignments. We end with some theoretical results when the demand streams are generated by MA(1). Here, we show that clusters resulting in the lowest MSFE are formed by grouping streams by the proximity of their MA coefficient.

The MA(1) case hints that in a general ARMA case, "best" clusters would be formed based on proximity of the ARMA coefficients between models generating the various streams (or equivalently based on the proximity of roots of the AR and MA polynomials).

Alternatively, best cluster assignments may result from grouping streams with most similar coefficients appearing in the MA($\infty$) representation. Future work can be performed to establish the best approach.

Our current theoretical approach based on general ARMA models is limited in that root-finding algorithms are unstable once the degree of a polynomial gets too large. In our study, this begins to occur when we consider the aggregate of around twenty streams with at least one AR coefficient. It is possible to greatly reduce the dimensionality, however, by first aggregating demand streams that are produced by identical or nearly-identical ARMA models. This is another direction for future research.

**Author Contributions:** All authors contributed equally to conceptualization, methodology, formal analysis and writing. All authors have read and agreed to the published version of the manuscript.

**Funding:** This research received no external funding.

**Data Availability Statement:** https://github.com/vkovtun84/Pivot-Clustering-of-Demand-Streams, accessed on 17 September 2023.

**Conflicts of Interest:** The authors declare no conflict of interest.

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
