# Peer review of "Pivot Clustering to Minimize Error in Forecasting Aggregated Demand Streams Each Following an Autoregressive Moving Average Model"

_stats, doi:10.3390/stats6040075_

Round 1
Reviewer 1 Report
Comments and Suggestions for Authors
Author Response
Thank you for your comments in helping us improve the paper! We look forward to checking the empirical implications of this research using real world datasets in future papers when we obtain the data and permissions to use it for research. In the meantime, as a proxy, we do carry out simulations where we use ARMA(1,1) models to produce demand realizations. ARMA(1,1) has been shown to be adequate in describing demand data such as appearing in Yu (2022), Hindawi, “Evaluation and Analysis of Electric Power in China Based on the ARMA Model”.
(1) We have adjusted the formatting as per your suggestions. We tabulated the definitions of several time-series sequences in 110-119 and simplified the notation in 149. We improved the structure of the formulas mentioned in (c) and (d).
(2) and (3)
We have defined BLF when it first appears and removed the extra parentheses in 221-223.
Thank you!
Reviewer 2 Report
Comments and Suggestions for Authors
The paper deals with a potentially interesting and relevant topic. The paper is clearly written. The derivation and presentation of the results is rigorous. My main issue is with the practical relevance of the results presented.
The paper is purely theoretical in nature. The results and the practical relevance of the results reported critically hinge on the assumptions about the demand-generating processes.
In this respect, I see a number of shortcomings which should be addressed. Observed demand time series typically exhibit non-stationary features e.g. seasonal patterns and trends. None of this is mentioned in the paper. The presence of such features may also have an effect on the results presented. Moreover, the authors use a wide range of stationary ARMA processes in various parts of the paper. E.g. the MA(1) processes in 2.5 with dependence parameters close to 1. In Section 3, AR and ARMA processes with persistence close to one. And finally numerically unspecified MA(1) processes in the later parts of the paper. Using these different processes makes me wonder if they are chosen specifically to drive home the various results and opens questions about the practical relevance of these types of processes.
Finally, the surprisingly low number of references (according to my counting only 3 (!) references are directly related to demand forecasting) calls into question the adequate embedding into the time series literature and its practical relevance.
Author Response
Thank you very much for your insights towards improving our paper. We focused on stationary ARMA models which are simple in order to keep the exposition as clear as possible. The results here extend to more complex models, but this will also lead to a more complex discussion. We do note that in the event that ARIMA models are considered, one would need to difference the ARIMA demand and ultimately obtain ARMA differences. The work carried out in this paper for ARMA would apply to this case as well.This discussion has been added on pages 4-5.
We have added a reference (Williams, 76) on the study of telephone data where ARIMA and Seasonal ARIMA models were considered. The authors demonstrated that sub-aggregated data can be beneficial compared to the aggregate when producing forecasts. Nonetheless, they only considered a few time-series models and did not attempt to determine how clusters should be selected. We leave to future research on the study of ARIMA and more complex seasonal time-series demand.
The example in Section 2.5 was chosen specifically to be simple enough to both explain the methods used in the theoretical MSFE evaluations in this paper and to highlight that subaggregated clusters can result in MSFE that is very close to the MSFE based on the individual streams. However as shown in section 4 there is a lot of benefit from finding clusters even when coefficients are randomly assigned.
In addition, on page 5, we have mentioned that ARMA(1,1) models (and others) can have coefficients which lead to seasonal patterns in demand realizations. Furthermore we added a reference that talks about ARMA(1,1) being a good model for some demand data (Yu 2022). We also now include a reference (Tabar 13) where forecasts are done for an aggregate of ARMA(1,1) demand sequences.
The example in Section 3 is used to highlight the usefulness of subaggregating (clustering) demand streams. In our simulations later, we consider random coefficients in our ARMA models which still show the benefit of clustering (in particular Pivot clustering) over using the aggregated series to forecast. The many simulations carried out show that the methodology is applicable generally and not only for a particular set of models. We have added some additional sentences on pages 17 and 22 to make this clearer.
The aim of Section 6 (where we consider MA(1) models) is to highlight the role that coefficients play in determining whether streams should be clustered together. We are able to provide several theorems in this respect since each demand model has only one coefficient in it. We have included a comment on this on page 37
Thank you for highlighting that we should add more references which we have done with four additional citations.
Reviewer 3 Report
Comments and Suggestions for Authors
The article is very well written, with the necessary care in notations, allowing the reader to follow the reading without getting confused.
Some interesting theoretical results were found, however I am still not convinced about the usefulness of this methodology. Some points that concern me:
1- The authors state that "Demand streams are assumed to follow autoregressive moving average (ARMA) processes". The article begins with an introductory line applied to a real problem, but the development is entirely theoretical and related to ARMA models. No application to real data is made. What is the reason? Without an application it seems to me that the introduction and the article are somewhat disconnected.
2- What is the ideal number of clusters? How to decide whether data should be clustered into 3 or 4 groups and how would this affect the methodology?
3- If one of the real time series has a unit root, which is quite common in time series, how should the user of the methodology proceed before aggregating the data?
4- Tables 4 and 5 are not very clear. Why not present it in terms of means and standard deviations?
5- I couldn't understand this step of the algorithm: "We then subaggregate the realizations to determine the subaggregated realizations for each cluster of demand streams and use these to estimate ARMA(5,5) models anytime it is necessary to obtain the ARMA model for a certain cluster and compute the in-sample forecast errors based on these realizations." Could the authors explain this step better?
6- I think the literature review is very poor. There are several articles that deal with linear combination of time series and forecasting properties. The authors need to expand this literature review.
Author Response
Thank you for your thorough evaluation of our paper! We believe it is now much improved.
- The idea was motivated from a real-life example, where we obtained data streams from 2000+ retail stores. We were able to observe the reduction in error from clustering. Unfortunately, this data is not available to reference. Instead, we simulated several instances to convince ourselves of the potential use of the proposed approach. Going forward, we will attempt to obtain field data that can be used. We also refer to Benjafaar et al. (2004) who cite several references related to demand allocation for planning purposes.
- We have added a couple of sentences to explain what the choice in the number of clusters is based on at the bottom of page 6.
- We have added a short discussion on ARIMA and SARIMA at the beginning of Section 2.
- We are not clear about this comment as there are no tables 4 and 5.
- We have improved the explanation in this paragraph.
- We have added a reference (Williams, 76) on the study of telephone data where ARIMA and Seasonal ARIMA models were considered. The authors demonstrated that subaggregated data can be beneficial compared to the aggregate when producing forecasts. Nonetheless, they only considered a few time-series models and did not attempt to determine how clusters should be selected. Furthermore we added a reference that talks about ARMA(1,1) being a good model for some demand data (Yu 2022) as well as (Tabar 13) who forecast an Aggregate of ARMA(1,1) demand sequences and added an additional reference to Benjaafar (2004) as mentioned.
Reviewer 4 Report
Comments and Suggestions for Authors
Please find my comments in the attached referee report.

N/A
Author Response
Thank you for your thorough insights to help improve this paper!
- We have cleared up these phrases where applicable.
- We have defined these under equation (1).
- We have simplified the notation in Section 2.1 and defined “hat” notation.
- We have provided an explicit calculation to the base case as requested. We removed the comment in the proof of Proposition 1 that St is ARMA for two ARMA processes X1 and X2 and leave that to the explanation underneath equation (22) on page 11 for why St is ARMA in the general case.
- We have added the explanation preceding {Dτ}τ=-∞t
6. We have fixed this throughout the document.
7. The claim at the top of Section 3 that the 10 given streams fall into three clusters of similar streams is really a grouping based on the orders of the ARMA models. The coefficients are chosen based on the intuition gleaned from the results of Section 6, where MA(1) streams are clustered based on the proximity of their MA(1) coefficients. We make this clearer at the top of the example. The natural cluster (based on type of ARMA model used) results in the lowest MSFE among various groupings mentioned in the table at the top of page 20. We do not have a unifying theme about proximity of coefficients in ARMA(p,q) models as there could be an infinite number of coefficients that need to be considered (in the MA(representation) for each stream. It is possible to assign a distance measure between two infinite series by considering the sum of squared differences between coefficients. This can be used as a direction for future study.
8. If streams are only considered once (ie. not moved a second time to other clusters) then the complexity is O(Nk) where N is the number of streams and k is the number of clusters. This is not considering MSFE calculation (either theoretical or estimated). We have added a mention to this in footnote 4.
9. We have added a new table (Figure 8) which displays how Pivot clustering performs against an exhaustive search which obtains the optimal clustering assignment. In this case we carry out twenty simulations based on assigning 10 streams to 3 clusters (using 10 different initializations of Pivot within each simulation). We observe that Pivot clustering typically finds either the global optimal solution or finds a grouping assignment whose MSFE is very close to that of the optimal solution.
10. In our simulations we considered what happens if we estimate the coefficients of the ARMA model that applies to each cluster based on the simulated data. We believe that looking into other possible data generation processes would be a good avenue for future research.
- We have corrected this throughout the document.
- We have added an explanation in the sentence.
- We have removed this sentence as the previous sentence explains what we are doing.
- We have defined BLF it in its first instance.
- We have fixed this sentence.
- It is now defined.
- We have fixed this sentence and moved it to a footnote.
Thank you! We feel the paper is now much improved!
Round 2
Reviewer 2 Report
Comments and Suggestions for Authors
()
Author Response
Thank you again for helping us improve the manuscript!
Reviewer 3 Report
Comments and Suggestions for Authors
Something went wrong when adding the new references. A small detail that needs to be revised.
Author Response
We are sorry, but we cannot find the issue. Could you please point to it in the revised manuscript (3rd submission). Thank you.
Reviewer 4 Report
Comments and Suggestions for Authors
I would like to thank the authors for taking the effort to revise the manuscript. There is one remaining issue pertaining to my comments from the previous round.
Specifically, the comment I had was "The entire analyses of the paper relies on the fact that the demand sequences follow exactly an ARMA model, which is rarely the case in reality. It would be good to see some sensitivity analysis, i.e., when the underlying true dynamics for the sequences deviate from ARMA, how the performance looks like."
This comment essentially pertains to settings where the data are generated according some dynamics that is DIFFERENT from an ARMA model, but an ARMR model is fit to the data (as an working model), followed by subsequently analyses. This is very doable in synthetic data experiments; additionally, given that the entire paper is built on the assumption that the demand sequence follows exactly an ARMA model (which we know does not hold in reality), such analyses provide insight into how useful the methodology is in practice.
The authors respond that "In our simulations we considered what happens if we estimate the coefficients of the ARMA model that applies to each cluster based on the simulated data. We believe that looking into other possible data generation processes would be a good avenue for future research."
I am not entirely sure what the response mean and how it is related to my comment.
Author Response
Thank you immensely for bringing this to our attention. We are sorry we did not adequately explore this in the previous revision. We have now added in new analysis (starting below Figure 8 and ending at the bottom of page 35). As per your suggestion, we have simulated data streams that are not generated by an ARMA model. Rather, each stream is generated using the ARFIMA(0,d,0) model where 'd' is selected randomly for each stream to be a value between -4 and .4. The results are quite interesting and help to highlight the value of finding good clusters (using an algorithm such as Pivot Clustering).
Computing the estimated MSFEs based on fitting ARMA models to the disaggregated, aggregated and subaggregated streams we see that it is actually possible for good clusters to outperform the disaggregated case (which is always optimal in the theoretical case). See Figure 9 for details.
Round 3
Reviewer 4 Report
Comments and Suggestions for Authors
I would like to thank the authors for their careful revision of the paper. At this point, I don't have further questions.